



**Atmospheric extremes triggered the biggest calving event in more than 50**
**years at the Amery Ice shelf in September 2019**
**Diana Francis [1*], Kyle S. Mattingly [2], Stef Lhermitte [3], Marouane Temimi [1], Petra Heil [4]**
[1] Khalifa University of Science and Technology, P. O. Box 54224, Abu Dhabi, United Arab
Emirates.
[2] Institute of Earth, Ocean, and Atmospheric Sciences, Rutgers University, New Brunswick, NJ,
USA.
[3] Department of Geoscience and Remote Sensing, Delft University of Technology, Mekelweg 5,
2628 CD Delft, Netherlands.
[4] University of Tasmania, Hobart, Tasmania 7001, Australia.
*Corresponding Author:* diana.francis@ku.ac.ae.
**Abstract**
Ice shelf instability is one of the main sources of uncertainty in Antarctica's contribution to future
sea level rise. Calving events play crucial role in ice shelf weakening but remain unpredictable and
their governing processes are still poorly understood. In this study, we analyze the unexpected
September 2019 calving event from the Amery Ice Shelf, the largest since 1963 and which
occurred almost a decade earlier than expected, to better understand the role of the atmosphere in
calving. We find that atmospheric extremes provided a deterministic role in this event. The calving
was triggered by the occurrence of a series of anomalously-deep and stationary explosive twin
polar cyclones over the Cooperation and Davis Seas which generated strong offshore winds
leading to increased sea ice removal, fracture amplification along the pre-existing rift, and
ultimately calving of the massive iceberg. The observed record-anomalous atmospheric conditions
were promoted by blocking ridges and Antarctic-wide anomalous poleward transport of heat and
moisture. Blocking highs helped in (i) directing moist and warm air masses towards the ice shelf
and in (ii) maintaining stationary the observed extreme cyclones at the front of the ice shelf for
several days. Accumulation of cold air over the ice sheet, due to the blocking highs, led to the
formation of an intense cold-high pressure over the ice sheet, which helped fuel sustained
anomalously-deep cyclones via increased baroclinicity. Our results stress the importance of
atmospheric extremes in ice shelf instability and the need to be accounted for when considering
Antarctic ice shelf variability and contribution to sea level, especially given that more of these
extremes are predicted under a warmer climate.





**Keywords:** Twin polar cyclones, explosive cyclones, blocking highs, ice shelf instability, calving,
East Antarctica, Amery Ice Shelf.

### 1. Introduction

The rapid collapse of several Antarctic ice shelves, observed recently, and the near-instantaneous
acceleration of land-ice discharge into the ocean that follows the collapse, demonstrates the
sensitivity of the Antarctic cryosphere to recent warming (e.g., Smith et al., 2019; Rignot et al.,
2019). However, large uncertainty remains regarding the response of ice shelves to the globally
rising temperatures and to the resulting changes in the atmospheric circulation.
On 25 September 2019, the Amery Ice Shelf – the third largest ice shelf in Antarctica – calved
iceberg D28 (1,636 km2, 210 m thick), which was the largest calving event since the early 1960s
(Fig. 1). The Amery Ice Shelf is a key drainage channel in East Antarctica (Fricker et al., 2002)
draining roughly 16% of the East Antarctic Ice Sheet (Galton-Fenzi et al., 2012). It is considered
in balance with its surroundings (King et al., 2009; Galton-Fenzi et al., 2012), despite experiencing
strong surface melt in summer. However, over the past 20 years, a large system of rifts (a precursor
to calving) in the Amery Ice Shelf, known as the Loose Tooth rift system, has been developing
(Fricker et al., 2005; Bassis et al., 2008). Recent studies have shown that the propagation rate of
the rifts has been decreasing since 2005 due to increasing thickness of melange ice filling in the
rifts, and speculated that forward propagation of the west rift might even stop (e.g., Zhao et al.,
2013). Satellite images of the Amery Ice Shelf (Fig. 1) show the largest rift extending in the same
direction of the ice flow, widening toward the edge of the ice shelf and from this main rift, with
radial rifts extending to the west (T1) and east (T2). Earlier studies predicted that the Amery Ice
Shelf would not experience a major calve until at least 2025 or later (e.g., Fricker et al., 2002), and
the portion that was expected to calve first was T2 i.e., the one to the east of the current calving.
This highlights the need for an improved understanding of the underlying processes of calving
events and the role of atmospheric forcing in ice shelf weakening; a precursor to rapid and major
changes in ice shelf stability.
Indeed, most of the mass loss from the Antarctic Ice Sheet – the largest uncertainty for future sea
level projections – takes place at the fronts of ice shelves and glacier tongues, via iceberg calving
and surface and basal melt (e.g., Pritchard et al., 2012; Shepherd et al., 2018). Compared to
melting, rifting and subsequent calving is the fastest way by which marine-terminating glaciers
lose mass to the ocean and contribute therefore to sea level rise (e.g., Smith et al., 2019). Despite
being floating ice (i.e., changes in their mass due to calving do not have a direct contribution to
sea level rise), ice shelves act to buttress inland ice by blocking the flow of ice from the interior.
This restrictive force decreases when ice shelves thin or calve. Calving from floating ice shelves
contributes indirectly to sea-level rise as these events accelerate the rate of ice flow from grounded
ice-sheet into the ocean (e.g., Hogg and Gudmundsson, 2017). For example, on the Antarctic
Peninsula, such events have been shown to increase by eight-fold the rate of ice flow inland
(Rignot et al., 2004; Scambos et al., 2004). This leads to more ice discharge into the oceans and a
consequent increase in the ice-sheet contribution to global sea-level rise (Hogg and Gudmundsson,
2017). Ocean-driven thinning was also detected at key ice shelves of the East Antarctic Ice Sheet
including the Amery Ice Shelf (Greenbaum et al., 2015; Smith et al., 2019) suggesting that this

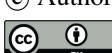



region is also susceptible to rapid and large-scale ice loss (Aitken et al., 2016), and could contribute
to future sea-level rise (DeConto et al., 2016; Rignot et al., 2019). Therefore, there is an urgent
need to assess the sensitivity of East Antarctic ice shelves to atmospheric forcing and to understand
the governing processes.
Beyond being part of a natural glaciological process, calving events at Antarctic ice shelves have
been attracting much attention recently (e.g., Liu et al., 2015; Benn and Astrom 2018) as they were
found to trigger, in some cases, the total disintegration of the parent ice shelf (Cook and Vaughan
2010; Liu et al., 2015; Jeong et al., 2016; Bassis and Ma, 2016; Massom et al., 2018). These events
have been attributed mainly to an enhanced regional warming (Vaughan et al., 2012; Pitchard et
al. 2012) which increases surface and basal melt as well as to ocean forcing involving intense
crevassing and rifting along multiple lines of weakness such as radial crevasses (Liu et al., 2015;
Jeong et al., 2016; Bassis and Ma, 2016), and to regional loss of pack ice in the shelf-front area
which allows storm-generated ocean swell to flex the outer margins of the shelves and lead to their
calving (Massom et al., 2018). However, atmospheric-dynamics forcing during such events,
particularly the wind mechanical action on rift widening both directly and via wind-induced waves,
remains unexplored.
Despite the importance and the implications of ice shelf calving, this phenomenon remains
unpredictable and is still poorly understood. Moreover, the underlying mechanisms governing
Antarctic ice shelf instability, especially those associated with atmospheric extremes, remains
unknown.
Of particular importance is the impact on Antarctic ice shelves of the poleward shift of
extratropical storm tracks (Tamarin and Kaspi, 2017) and the observed increase in the number and
intensity of cyclones around Antarctica over the last few decades (Wei and Qin 2016). The
poleward shift of extratropical cyclones was found in reanalysis data of recent years (Fyfe, 2003;
Son et al., 2008), and models project an estimated poleward shift of cyclone genesis 1° to 2° in
latitude on average under enhanced greenhouse gas concentrations (Bengtsson et al., 2009; Barnes
and Polvani, 2013). Importantly, this poleward shift was found to be particularly pronounced in
the Southern Hemisphere (Chang et al., 2012), and the mean intensity of cyclones as well as the
number of extreme cyclones are projected to increase under a warmer climate scenario (Lambert
and Fyfe, 2006; Chang, 2017; Kossin et al., 2020).
Changes in cyclone tracks, numbers, and intensity may have significant impacts on Antarctic sea
ice and land ice. In fact, weather systems (i.e., cyclones and blocks) resulting from the larger-scale
circulation (e.g., Pope et al., 2017) are identified as the main driver of the observed trends in sea
ice variability (Matear et al., 2015; Schemm, 2018; Turner et al., 2017; Eayrs et al., 2019).
Furthermore, cyclones and their associated atmospheric rivers can induce sea ice melt (Francis et
al., 2020), ice-shelf surface melt (Wille et al., 2019) and significant sea ice drift (Kwok et al., 2017;
Francis et al., 2019) by virtue of their anomalous moisture and heat transport to high latitudes
(Woods & Caballero, 2016; Grieger et al., 2018) and the strong surface winds they carry (Schemm,
2018). Severe storms can generate energetic waves (up to 8 m) in the Southern Ocean capable of
penetrating hundreds of kilometers into the sea ice covered ocean (Kohout et al., 2014; Vichi et
al., 2019). Concomitantly, the sea ice cover acts as a buffer and attenuates the wave energy over





distance, reducing therefore the impact of storms on ice shelves (Dolatshah et al., 2018; Massom
et al., 2018).
An extreme situation in cyclogenesis is the formation of explosive cyclones. These are developing
cyclones for which the central pressure decreases by at least 24 hPa in 24 hours (Sanders and
Gyakum, 1980). Explosively developing cyclones are deeper and longer-lasting compared to
ordinary cyclones and they are found to be more intense in the Southern Hemisphere than in the
Northern Hemisphere (Raele et al., 2019). In particular, explosive cyclones in the Indian Ocean
sector of the Southern Ocean (close to South Africa) are stronger and express higher deepening
rates than elsewhere around Antarctica (Raele et al., 2019). This same region (between 45°E and
90°E and poleward of 40°S) – encompassing the Amery Basin – stands out in a climatological
study (Allen et al., 2010) as one of three main regions for explosive cyclogenesis around
Antarctica, where explosive cyclones are characterized by a 20hPa mean pressure depth relative
to the surrounding pressure field. A climatological study of explosive cyclones (Lim and
Simmonds, 2002) found that the number of explosive cyclones increased in both hemispheres
during 1979-1999, and that positive trends of such systems are statistically significant in the
Southern Hemisphere. On average, the study identified 26 explosive cyclones per year in the
Southern Hemisphere and found that explosive cyclones exhibit greater mean intensity and depth
relative to the entire population of ordinary cyclonic systems. A more recent climatological study
over a longer period (1979–2013) reported similar findings, with an increase in the frequency of
explosive cyclones in the band of 45°–55°S during winter and early spring (Wei and Qin 2016).
The spatial distribution of these cyclones was found to have a close association with that of strong
baroclinicity. In general, the preferred region for cyclogenesis is where both a strong temperature
gradient and an upper-level trough are present (e.g., Shimada et al., 2014). While high baroclinic
instability associated with the horizontal temperature gradient is crucial for the formation and the
intensification of cyclones (Davies, 1997, Uccellini, 1990), cyclogenesis occurs only at the
entrance and exit regions of upper-level troughs (e.g., Shimada et al., 2014). Around Antarctica,
the strongest temperature gradient is found during late winter-early spring along the fringes of the
ice pack, making the sea-ice edge a preferred region for cyclogenesis (e.g., Schlosser et al., 2011;
Stoll et al., 2018). However, the location of the temperature gradient relative to the ice edge
depends strongly on the atmospheric circulation at larger scale, where a strong temperature
gradient can occur poleward of the ice edge (i.e., closer to the ice shelves) during an enhanced
zonal wave number three (ZW3) pattern (Francis et al., 2019). This pattern is characterized by the
alternation of 3 troughs and 3 ridges around Antarctica. Strong poleward transport of heat and
moisture occurs in the ascending branch of troughs and strong equatorward transport of cold air
occurs in the descending branch of ridges (e.g., Raphael, 2007). This zonally-alternating pattern
of cold and warm air masses creates temperature differences between the different sectors, fuels
frontogenesis and promotes the development of explosive cyclones close to the ice shelves and
over the sea ice cover.
Another aspect of the ZW3 pattern is the impact of the ridges on the propagation speed of the
cyclones. In the troughs, the extratropical cyclones and the associated moisture and heat fluxes are
directed poleward; once they reach the Antarctic coast they are blocked by the ridges to their east



(Francis et al., 2019; 2020). This results in stationary cyclones over the same region for 1-2 days
which in turn induces pronounced impact on the sea ice (e.g., Francis et al., 2019) and waves
(Vichi et al., 2019). The same scenario can happen at the front of ice shelves during winter-spring
if the cyclones form closer to the coast and/or the sea ice extent decreases under a warmer climate.
Interestingly, the Antarctic sea ice extent has been decreasing since 2015 (Swart et al., 2018) and
the ZW3 index has been the most positive on record during the same period (Schlosser et al., 2018;
Francis et al., 2019). Increased warm air advection toward Antarctica was found to be at the origin
of the observed negative anomaly in Antarctic sea ice extent in recent years (Schlosser et al., 2018).
Given the dual impact of ZW3 circulation on both explosive cyclogenesis (location and intensity)
and sea ice extent, this combination may result in a more pronounced impact of extreme cyclones
on ice shelves.
Another extreme situation in cyclogenesis is the formation of twin cyclones during which the
resulting effect of the mutually-interacting cyclones is twice as strong as the individual cyclones
(e.g., Moustaoui et al., 2002). To our knowledge, the formation of explosively developing twin
cyclones has been, to date, only observed and studied in the tropics (Ferreira et al., 1996;
Moustaoui et al., 2002), in the mid-latitudes (Yokoyama and Yamamoto, 2019) and in the Arctic
(Renfrew et al., 1997). In this study, we report for the time, the formation of polar twin cyclones
near Antarctica during two consecutive events; one on 19-20 September 2019 at 60°E and the
second on 23-24 September 2019 at 85°E.
Despite the observed poleward shift of extratropical cyclones, the increasing number and intensity
of explosive cyclones around Antarctica and the decline in sea ice extent in recent years, the impact
of extreme cyclones on ice shelves instability has not been investigated to date. This is the
objective of this study.
Building on previous studies that investigated these patterns separately, we aim in this study to
assess the impact of extreme cyclone activity during the largest calving event since 1963 at the
Amery Ice Shelf. Using satellite data and atmospheric reanalyses, we investigate the role of
atmospheric forcings in this calving event which occurred under a ZW3-like situation. The
development of the explosive cyclones and their impact on sea ice and land ice conditions are
addressed in section 2. Section 3 discusses our findings. The data and methods used in this study
are described in section 4.

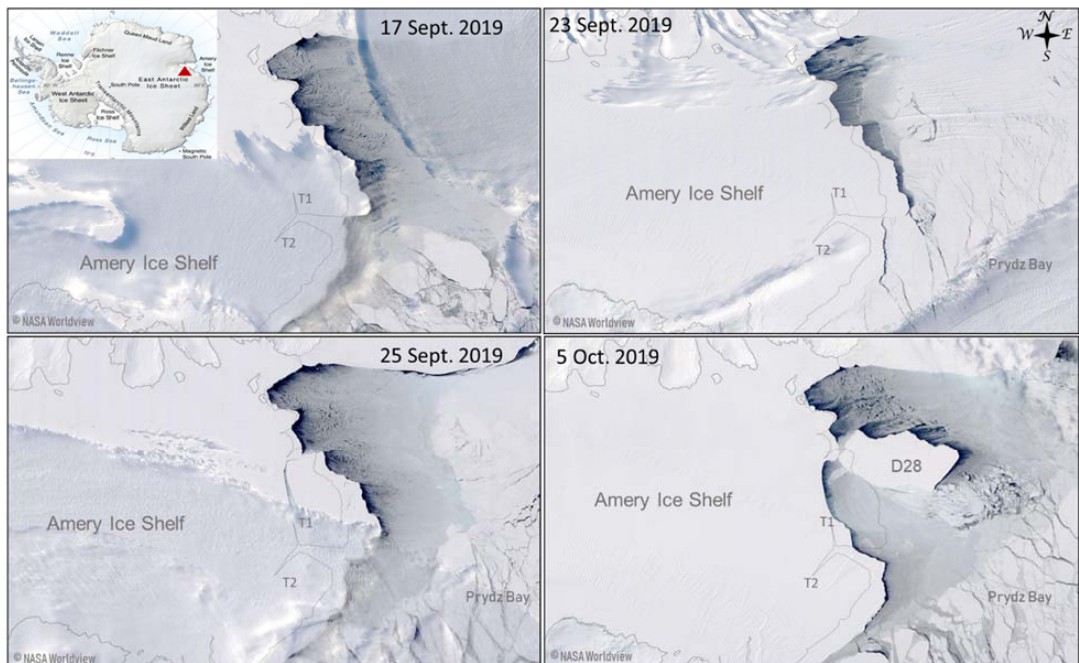

**Figure 1:** MODIS satellite visible imagery of the Amery Ice Shelf and the Loose Tooth rift system (T1 and T2) at its front. Ice conditions are shown before the calving on 17 and 23 September 2019, during the calving on 25 September 2019, and few days after the detachment of the new iceberg D28. Image credit NASA Worldview.


## 2. Results

### 2.1 Explosive twin cyclones during 18-22 September 2019 – preconditioning

In September 2019, the synoptic conditions exhibited an amplified zonal wave number 3 (ZW3) pattern characterized by 3 trough/ridge systems associated with low/high mean sea level pressure (MSLP) anomalies. Compared to all Septembers in the 1979-2019 period, the broad scale MSLP anomaly indicates that, for September 2019, there was below average pressure over much of the Antarctic continent and above average pressure to the north (Fig. 2a). In the Indian Ocean sector, the MSLP anomalies exceeded one standard deviation from the mean over large areas with the strongest troughing over Cooperation and Davis Seas (Fig. 2a). To the west of this low pressure anomaly, the South Atlantic ridge exhibited strong positive anomalies exceeding 2 standard deviations from the mean (Fig. 2a). To the east of the low pressure anomaly around the Amery Ice Shelf, another pronounced ridge encompassing south Australia and the Mawson Sea with positive MSLP anomalies exceeded 1 standard deviation from the climatological mean (Fig. 2a).

On a daily scale, the aforementioned synoptic setting was synonym of frequent and extreme weather systems. On 17 September 2019 at 0200 UTC, an extratropical cyclone associated with a 968 hPa low-pressure at its center and located at 60°S, 40°E, started to deepen while moving poleward and eastward. It reached the western side of Cooperation Sea on 18 September 2019 at



0200 UTC with a 940 hPa minimum pressure and remained over this region the entire day (Fig.
2b), then decayed on 19 September 2019 at 1300 UTC. The rapid deepening of the low pressure
is characteristic of explosive cyclones (e.g., Sanders and Gyakum, 1980). The explosive cyclone
on 18 September 2019 was associated with significant poleward transport of moisture (Fig. 2c)
and heat (Fig. 2d) carried by an atmospheric river propagating poleward adjacent to the low-
pressure center. The atmospheric river was associated with integrated water vapor transport (IVT)
greater than 500 kg m$^{-1}$ s$^{-1}$ at its core, with IVT values around 100 kg m$^{-1}$ s$^{-1}$ over Prydz Bay
exceeding the 99th percentile of September climatology in this region (Fig. 3a).

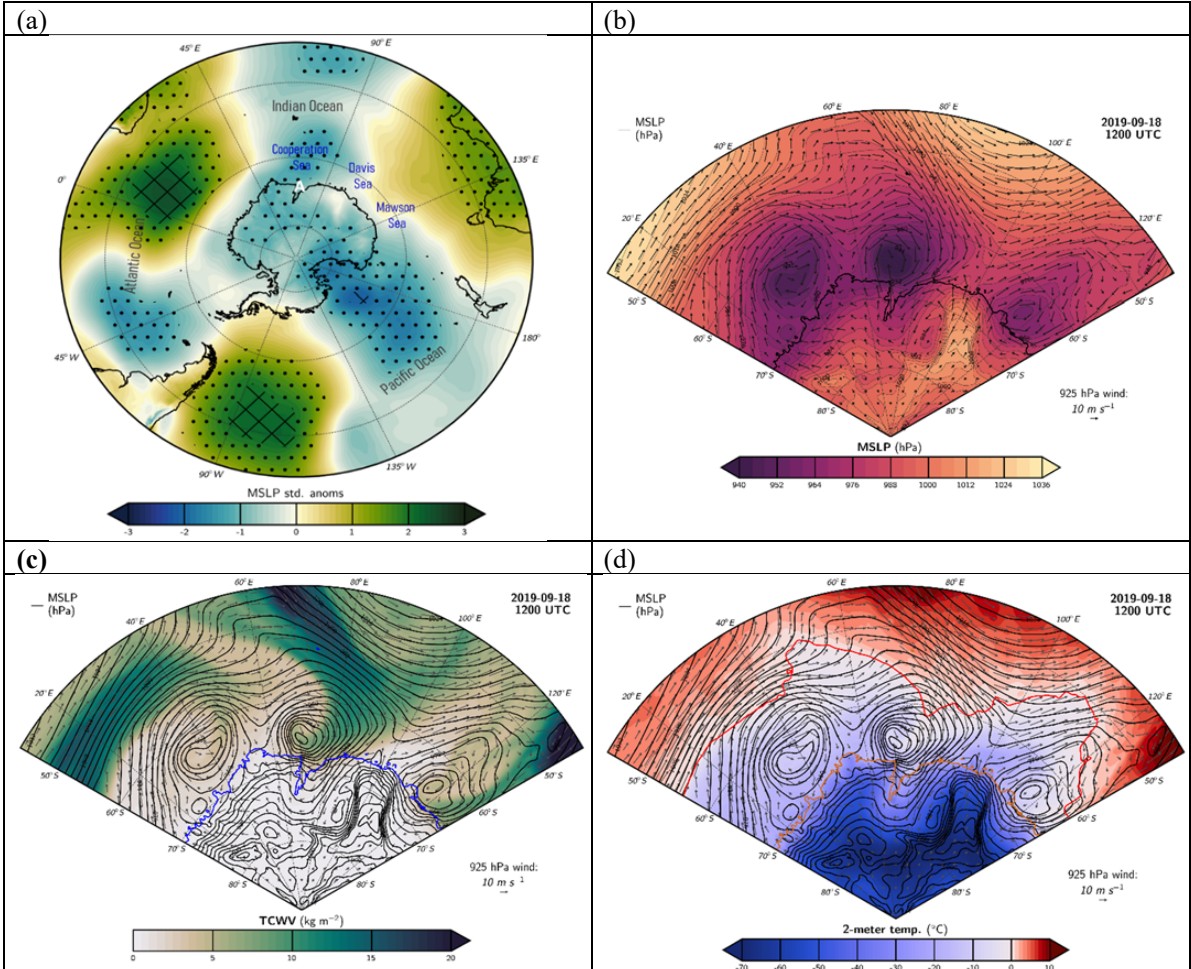

**Figure 2:** Normalized anomalies of Mean Sea Level Pressure (MSLP) for September 2019 relative to the
1979-2019 September climatology. Black dots are regions where the normalized anomalies are larger than
1 standard deviation from the mean and black squares are regions where the normalized anomalies are
larger than 2 standard deviations from the mean. The letter A in white indicates the location of the Amery
Ice Shelf. (b) MSLP (shaded) and winds at 925hPa (vectors) at 18 September 2019 1200 UTC, (c) same
as (b) but for the total column water vapor (TCWV) in colors, winds at 925hPa in vectors and MSLP in



black contours, (d) same as (c) but for 2-m temperature (in colors), winds at 925hPa in vectors, MSLP in black contours and 0°C contour in red.


In the Southern Hemisphere, where cyclonic winds spin clockwise, the highest wind speed occurs
along the bent-back front of the cyclone, i.e., to the left of the low-pressure center of the cyclone
(e.g., Wagner et al., 2011, Watanabe and Niino, 2014). This was observed during the explosive
cyclone on 18 September 2019 which generated extremely strong surface winds to the left of its
center exceeding 20 m s$^{-1}$ (Fig. 3b). Being stationary over Cooperation Sea but to the west of the
Amery Basin, this extreme cyclone generated a sustained northeasterly wind stress over the
northern part of the ice shelf (Fig. 3b), as well as strong poleward warm and moist air advection
(Fig. 2c, 2d and 3a). The combination of warm temperatures brought by the cyclone/AR and strong
easterly/northeasterly wind speeds was unusual (Fig. 3). MSLP anomalies during this event were
in excess of -4 sigma (Fig. 3b), with MSLP values below the 1st percentile of September
climatology over a large area along and to the north of the ice shelf margin (Fig. 3b). Extreme
wind anomalies exceeding the 99th percentile over the central and eastern ice shelf margin were
associated with this cyclone from 18 September through 19 September 2019 (Fig. 3c). Likewise,
there were sustained positive 2-m temperature anomalies throughout the period exceeding 2
standard deviations from the climatological mean (Fig. 3d).

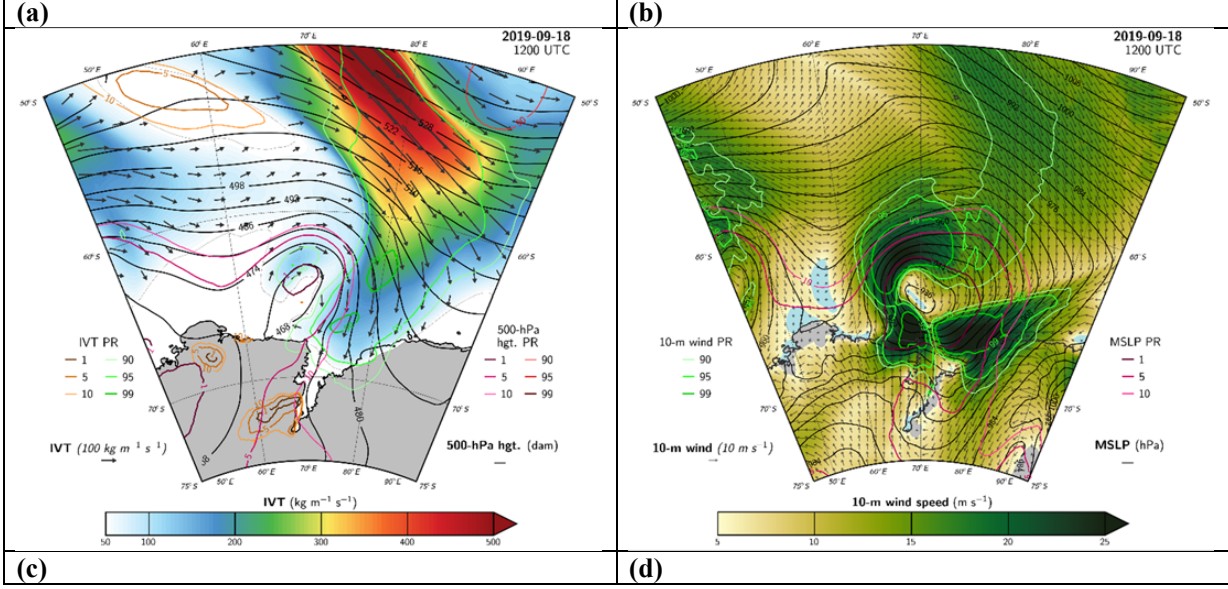

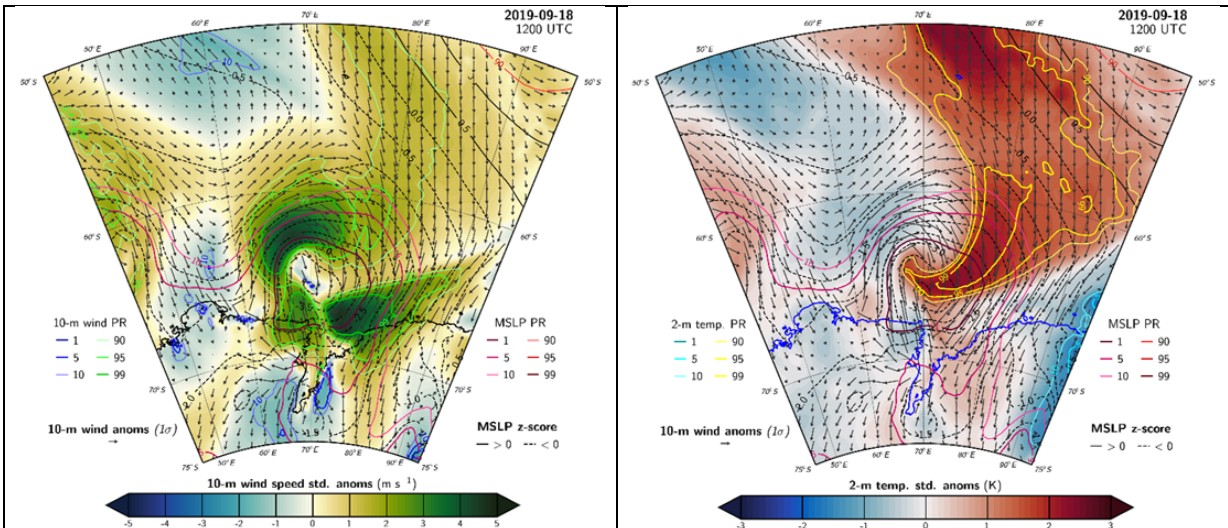

**Figure 3:** Maps on 18 September 2019 at 1200 UTC of (a) Integrated water vapor transport (IVT) shaded, geopotential heights at 500 hPa in black contours and IVT direction in black vectors, (b) 10-m wind speed in colors, 10-m wind direction in black vectors and MSLP in black contours. (c) Standardized 10-m wind speed anomalies relative to the full September record (1979-2019) (d) Same as (c) but for 2-m temperature. Colored contour lines show percentile rank extremes (1, 5, 10 and 90, 95, 99 percentile ranks) of the corresponding quantities indicated on the plots. On (c) and (d): Vectors show 10-m wind anomalies, black contours show positive MSLP anomalies and dashed black contours show negative MSLP anomalies.


The first explosive cyclone on 18 September 2019 was followed immediately by a second
explosive cyclone which approached Cooperation Sea from the west on 19 September at 1400
UTC with a deep low of 952hPa. At 2000UTC, this deep cyclone widened and evolved into two
twin polar cyclones over the same region (Fig. 4a). The twin cyclones exhibited 960hPa low-
pressure at their respective centers and remained active to the west of the Amery Ice Shelf for three
consecutive days (Fig. 4a). Their signatures dissipated in the pressure field on 22 September 2019
at 0000UTC. The poleward transport of heat (Fig. 4b) and moisture (Fig. 4c) towards the Amery
Ice Shelf continued during this event together with extreme wind stress exceeding the 99th
percentile (Fig. 4d). Being stationary to the west of the Amery Ice Shelf (Fig. 4a), the twin cyclones
induced extreme easterly winds across the ice shelf, with u-wind anomalies exceeding -5 sigma of
September climatology over the western ice shelf from 19 September 2019 at 1900 UTC through
20 September at 1100 UTC (Fig. 4d) and below the 1 percentile u-wind values over the whole
lower ice shelf area (Fig. 4d). When compared with the climatology for all months during 1979-
2019, many hourly wind speeds over the ice shelf front during 18-20 September were substantially
greater than the 99th percentile of climatology, with the most anomalous wind speeds on 18
September (Fig. 6e).
On 21 September 2019, the twin cyclones merged and moved to the area in front of the Amery Ice
Shelf (Fig. 4e) resulting in a deep cyclone associated with MSLP at its center below the 5[th]
percentile. The remnant cyclone slowly meandered along the northern margin of Prydz Bay and





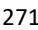

decayed on 22 September 2019. Anomalously warm air masses were brought by this cyclone over
the margins of the Amery Ice Shelf exceeding the 90th percentile (Fig. 4e). MODIS satellite
imagery on this day showed a swirling cyclone at the mouth of the Amery Ice Shelf (Fig. 4f).
Sentinel-3A and 3B observations on 22 September 2019 at 0000 UTC (i.e., during the decay of
the cyclones) show elevated waves at the ice-shelf front area reaching 6 m significant wave height
(Fig. 4g). Waves generated by the cyclones during the 18-21 September 2019 period, when easterly
wind speeds were stronger, may have been substantially higher. The easterly direction of the winds
during this episode infers that the wind-induced wave at the ice shelf front occurred likely in a
direction parallel to the pre-existing rift T1 at the western side of the front.
Surface melt during this event may have occurred briefly due to the anomalous warm and moist
air masses. However, the inspection of daily satellite images of Sentinel-1 backscatter coefficient,
MODIS ice surface temperature and AMSR2 brightness temperature did not show any prolonged
nor significant surface melt at the Amery Ice Shelf during this event.
In summary, an extended period of strong cyclonic activity from 18-22 September 2019 resulted
in exceptional period of strong easterly / northeasterly winds over the western side of the Amery
Ice Shelf where the climatology shows a positive zonal component. This exceptional wind stress
on the ice shelf generated strong waves in the region in front of the ice shelf. The advection of
anomalous warm and moist air masses to the area at the ice shelf front may have contributed to a
decrease in sea ice concentration at the front of the ice shelf, as it will be shown in section 5.

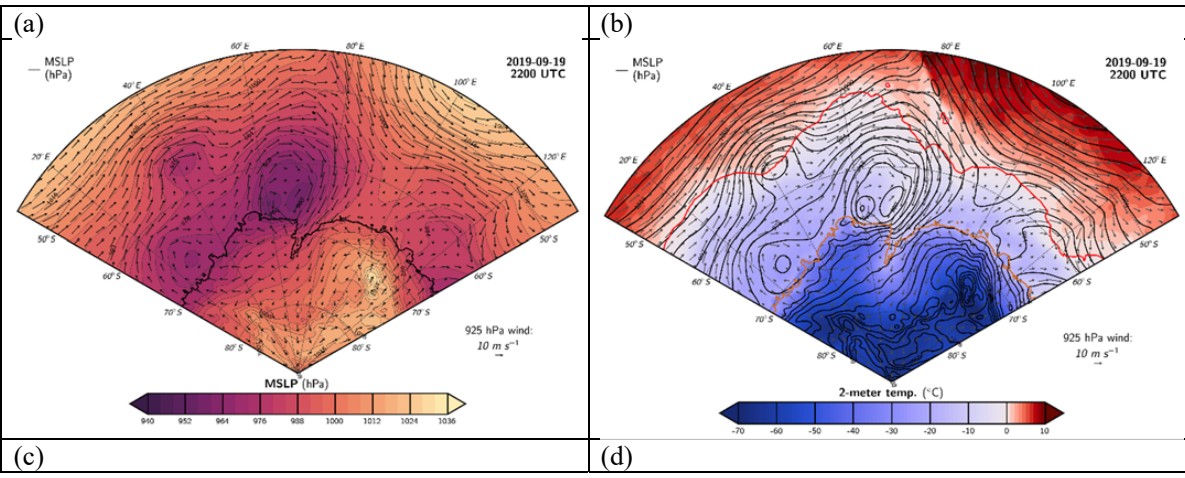







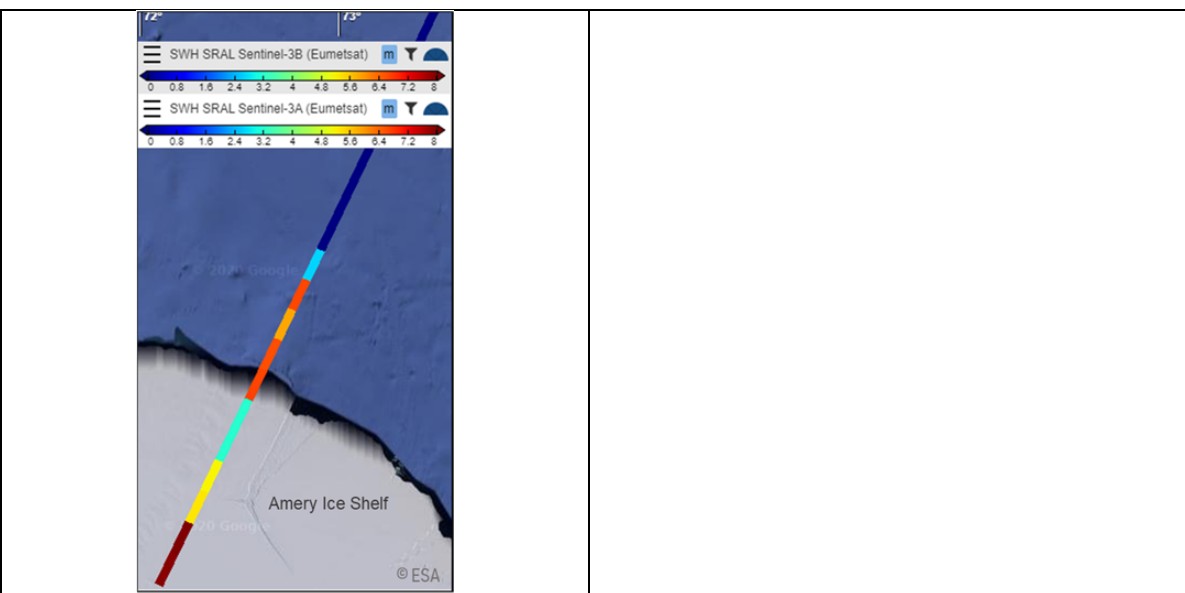

**Figure 4:** ERA5 reanalysis of: (a) MSLP in colors and winds at 925hPa in vectors on 19 September 2019 at 2200 UTC, (b) 2-m temperature in colors, winds at 925hPa in vectors, MSLP in black contours and 0°C contour in red on 19 September at 2200 UTC, (c) total column water vapor (TCWV) in colors, winds at 925hPa in vectors and MSLP in black contours on 20 September 2019 at 0000 UTC, (d) standardized anomalies relative to the full record (1979-2019) of 10-m u-wind on 20 September 2019 at 0000 UTC. Vectors show 10-m wind anomalies, black contours show positive MSLP anomalies and dashed black contours show negative MSLP anomalies. Colored contour lines show percentile rank extremes (1, 5, 10 and 90, 95, 99 percentile ranks) of the corresponding quantities indicated on the plots, (e) 2-m temperature in colors, 10-m winds in vectors, MSLP in black contours and 0°C contour in red dashed-line on 21 September at 0600 UTC. (f) MODIS visible imagery on 21 September 2019, image credit: NASA worldview. (g) Sentinel-3A and 3B observations of wave height on 22 September 2019 at 0000 UTC, image credit: ESA Ocean Virtual Laboratory.


## 2.2. Twin polar cyclones during 23-24 September 2019 - calving

Following the extended period of extreme cyclones in Cooperation sea, an explosive cyclone started to develop on 21 September 2019 centered at 45°E and 60°S. The pressure at its center deepened from 976hPa on 21 September at 1900 UTC to 952hPa on 22 September 2019 at 19 UTC (not shown). On 23 September 2019, the large explosive cyclone entered Cooperation Sea from the west with a deep low of 940 hPa (Fig. 5a). It was accompanied by an intense atmospheric river exhibiting core IVT greater than 800 kg m$^{-1}$ s$^{-1}$ and stretching from mid-latitudes towards Antarctica (Fig. 5b). The explosive cyclone was stationary over Cooperation Sea during the whole day on 23 September 2019, being trapped between two far-south reaching blocking ridges one to the west of it and the second to its east (Fig. 5a and 5c). The cyclone intensified, increased in size and evolved into twin cyclones on 24 September 2019 at 0000 UTC associated with 952hPa low pressure at their respective centers (Fig. 5c and 5d). The mutual interaction between the two cyclones appeared as co-rotation and an eastward translation of the binary pair by the ambient



flow. The interplay between the cyclones lasted for one day after which the twins merged and
decayed on 25 September 2019.
To the south of the twin cyclones, a cold pressure high (1036 hPa) developed over the ice sheet as
a result of the accumulation of cold air due the blocking ridge to the northeast (Fig. 5c). The high-
pressure advected extremely cold air masses (2-m temperature below -40°C) into the twin-cyclone
system (Fig. 5e and 5 f) which may have fostered baroclinicity and frontogenesis, hence sustaining
the twin cyclones for a longer period of time.
Transport of moisture from mid-latitudes toward East Antarctica continued during this period and
high precipitable water amounts were continuously advected by the atmospheric river and the
cyclones over the ice shelf margins (Fig. 5d). Sustained advection of exceptionally warm air
masses was observed during this event as well (Fig. 5e and 5f). Air masses characterized by 0°C
2-m temperatures were seen to penetrate further south reaching 66°S over the region to the east of
the twin cyclones during the whole day on 24 September 2019 (Fig. 5e and 5f).

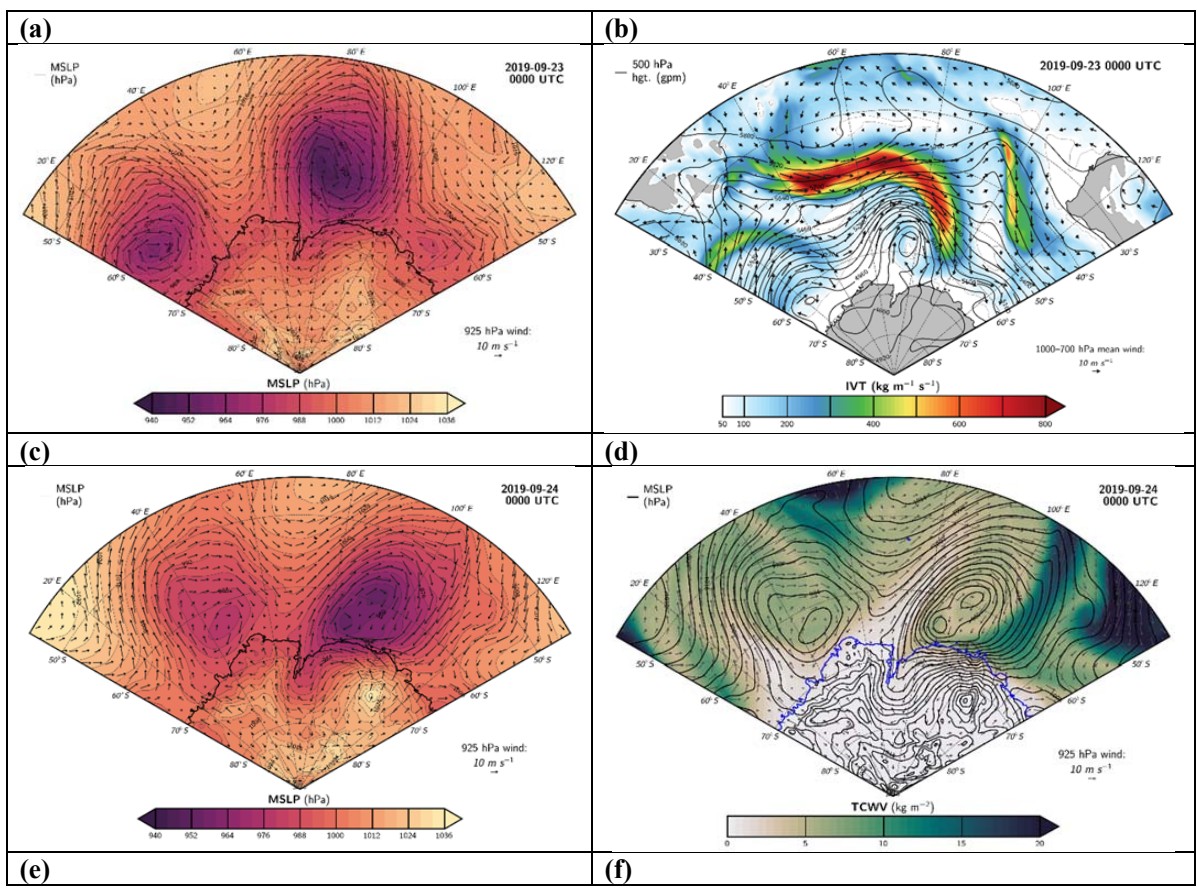

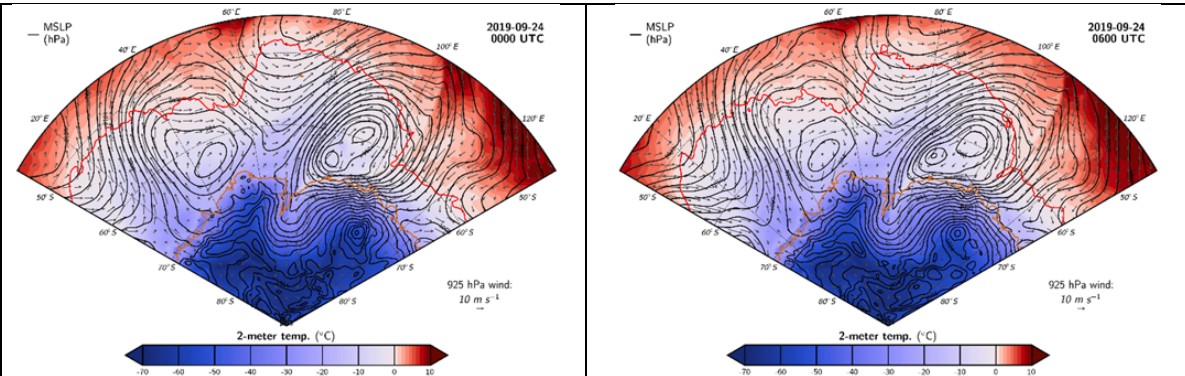

**Figure 5:** (a) MSLP in colors and winds at 925hPa in vectors on 23 September 2019 at 0000 UTC, (b) integrated water vapor transport (IVT) in colors, geopotential heights at 500 hPa in black contours and 1000-700 hPa mean winds in black vectors on 23 September 2019 at 0000 UTC, (c) total column water vapor (TCWV) in colors, winds at 925hPa in vectors and MSLP in black contours on 24 September 2019 at 0000 UTC, (d) MSLP in colors and winds at 925hPa in vectors on 24 September 2019 at 0000 UTC, (e) 2-m temperature in colors, winds at 925hPa in vectors, MSLP in black contours and 0°C contour in red on 24 September at 0000 UTC, (f) same as (e) but at 0600 UTC.


During 23-24 September, the deep twin polar cyclones were stationary to the east of the Amery
Ice Shelf (Fig. 5) associated with MSLP anomalies at their centers below the 5th percentile (Fig.
6a and 6c). They induced extreme westerlies (10-m wind speed in the order of 17 m s⁻¹) across the
ice shelf with positive 10-m wind anomalies exceeding 2 standard deviations from the
climatological mean (Fig. 6a). The direction of the winds was also exceptional with above 99th
percentile u-wind over the western ice shelf margin from 23 September 2019 at 1800 UTC (Fig.
6b) through 24 September 2019 at 1200 UTC and below the 5th percentile u-wind values over the
lower eastern ice shelf area (Fig. 6a and 6b). Weaker but still significant (95 percentile) westerly
wind anomalies lingered during the remainder of the day on 24 September 2019 and through
midday on 25 September 2019 with wind speed at 10-m reaching 15 m s⁻¹ at the front of the ice
shelf (Fig. 6d). Sustained positive 2-m temperature anomalies were observed throughout the twin
cyclone event over the eastern side of Prydz Bay. Warm air advection by the twin cyclones brought
percentile rank temperatures over the eastern side of the Amery Ice Shelf and Prydz Bay on 23
and 24 September 2019 and 90 percentile rank temperatures inland over Princess Elizabeth Land
(Fig. 6c). These episodes of poleward advection of warm air masses may explain the observed
positive-trend in surface temperatures during winter/spring seasons at Prydz Bay reported by Heil
(2006) using measurements from ground stations.
The distribution of hourly 10-m wind speed for all months 1979-2019 over the Amery Ice Shelf
front is shown in the histogram in Fig. 6f. The winds during the 18-22 September 2019 period
were exceptionally unusual compared to the record. The winds during the 23-25 September 2019
period were strong but not unusually extreme. This suggests that the first extreme cyclones' event
had an important role in preconditioning the ice shelf front for breakoff, while the offshore winds





during the second event triggered the calving by pushing the iceberg-to-be out from the shelf along
the T1 rift.



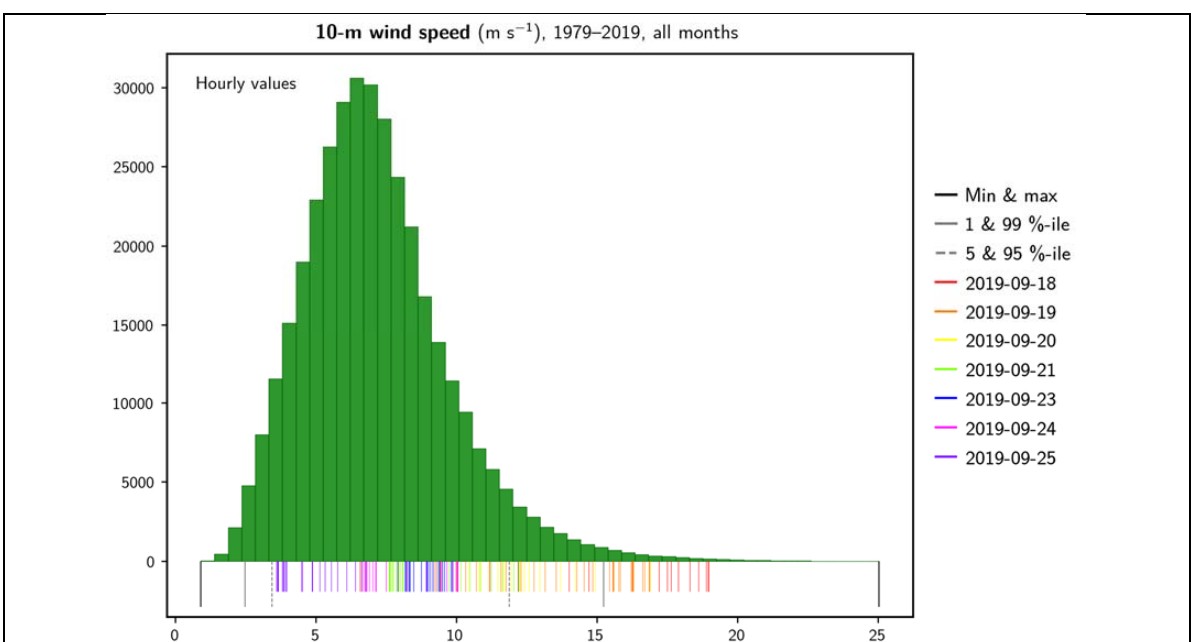

**Figure 6:** (a) Standardized anomalies relative to the full September record (1979-2019) of 10-m wind speed (colors) at 23 September 2019 1800 UTC. (b) Same as (a) but with u-component of 10-m wind as filled contours at 24 September 2019 0000 UTC. Vectors show 10-m wind anomalies, black contours show positive MSLP anomalies and dashed black contours show negative MSLP anomalies. Colored contour lines show percentile rank extremes (1, 5, 10 and 90, 95, 99 percentile ranks) of the corresponding quantities indicated on the plots. (c) 2-m temperature in colors, winds at 925hPa in vectors, MSLP in black contours and 0°C contour in red dashed-line on 23 September at 2100 UTC. (d) 10-m wind speed in colors, 10-m wind direction in black vectors and MSLP in black contours on 24 September 2019 at 1000 UTC. (e) Histogram showing the distribution of hourly 10-m wind speed for all months during 1979-2019, spatially averaged over 70-65°S and 70-75°E. The colored vertical lines correspond to hourly values during the 18-25 September 2019 period. Each given day in September 2019 has 24 hourly values plotted in the same color.


### 2.3 Sea ice and land ice conditions

The anomalous atmospheric conditions during the extended period of strong cyclonic activity
occurring over the ice cover in Cooperation Sea (i.e., south to the sea ice edge) had significant
impacts on both sea ice and land ice (Fig. 7). The sustained period of strong cyclonic activity
occurring over the sea ice pack and onto the Amery Ice Shelf caused decreases in sea ice
concentration both at the mouth of the Amery Ice Shelf and further offshore. Although this study
focuses on the period 17-25 September 2019, the inspection of the sequence of MODIS images for
the whole month of September 2019 revealed several episodes of sea ice removal from the ice
shelf front area during the 7-17 September 2019 period by offshore winds (i.e., Fig. 1). Despite
the formation of new sea ice over the area, the sea ice removal may have preconditioned the sea




ice cover for further reduction during the subsequent series of extremes cyclones and increased the
area of open water susceptible to ocean wave activity along the front of the ice shelf.
Sea ice concentration in Cooperation Sea and at the Amery Ice Shelf front area was reduced to
below 60%, reaching 40% in some places (Fig. 7a and 7b). By the end of the intense cyclonic
activity period, areas of open water formed especially at the locations of the strongest surface
winds i.e., to the left of the twin cyclones centers (Fig. 7c and 7d). Significant reduction in sea ice
concentration was also observed along the sea ice edge associated with wind-driven currents and
waves (Fig. 7) which may have decreased the sea-ice attenuation effect of waves-in-ice
propagating from lower-latitude ocean toward the ice shelf.
The decrease in sea ice concentration was due to both sea ice melt caused by the anomalous warm
and moist air masses advected over the Amery Ice Shelf during the first episode of twin cyclones
(i.e., Fig. 3 and 4) and to sea ice drift out of the region by strong winds during the second episode
of twin polar cyclones (i.e., Fig. 5 and 6). The strong waves generated locally in the area of reduced
sea ice concentration in front of the ice shelf during the first set of cyclone events (Fig. 4g), were
important in preconditioning the breakoff by inducing flexure at the front.
Significant sea ice drift was observed at the mouth of the Amery Ice Shelf associated with the
exceptional westerlies generated by the twin cyclones on 23-24 September 2019 (Fig. 7e and 7d).
The sea ice drift velocity during this period reached 50 km on average per day and the sea ice
drifted away from the Amery Ice Shelf towards the east and northeast (Fig. 7f).

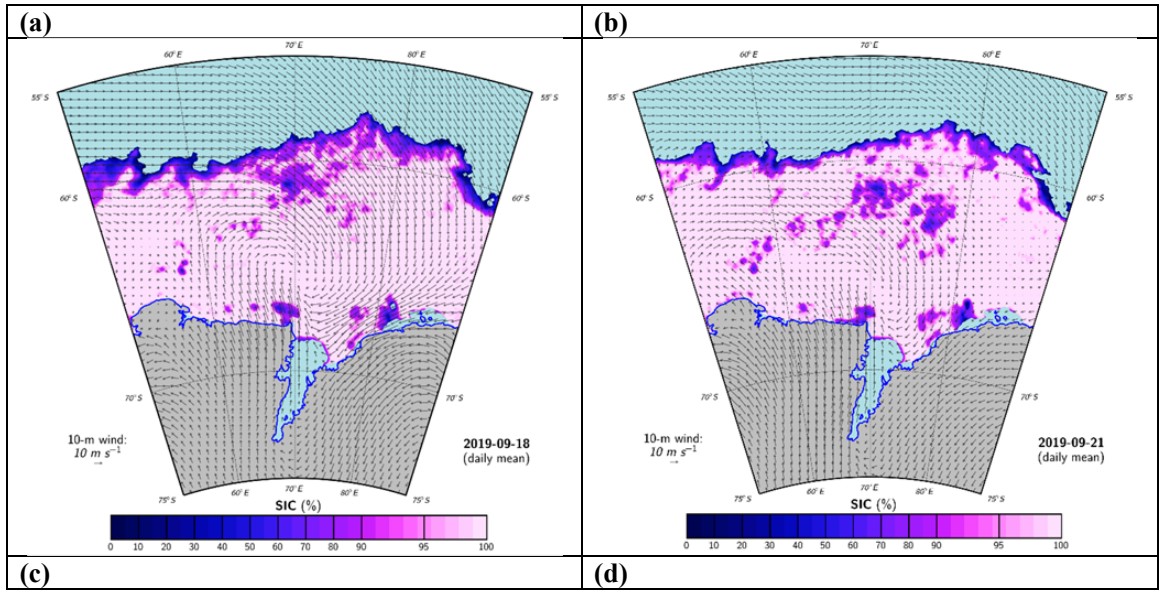











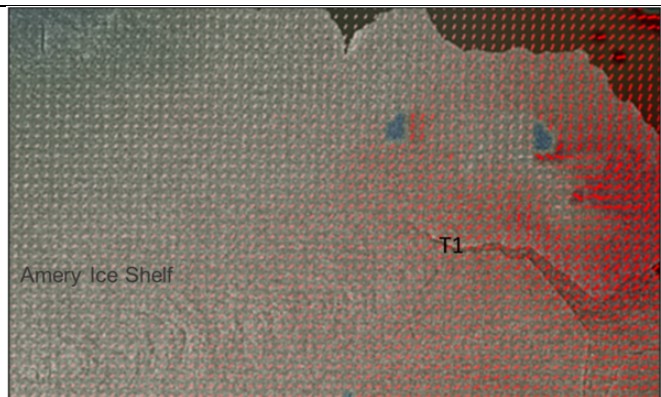

**Figure 7:** Satellite-derived sea ice concentrations and ERA5-derived daily mean 10-m winds in vectors over 55-75S and 50-90E on: (a) 18 September 2019 at 1200 UTC, (b) on 21 September 2019 at 1200 UTC, (c) 23 September 2019 at 0000 UTC and (d) on 24 September 2019 at 0000 UTC. Satellite-derived daily sea-ice drift velocity in colors and direction in vectors on 23 September 2019 (e) and 24 September 2019 (f). The solid pink contour is the 0% sea ice concentration contour and the solid purple contour is the 95% sea ice concentration contour. (g) SAR image of the Amery Ice Shelf on 23 September 2019. The red vectors correspond to the ice displacement on 23 September 2019 (both velocity and direction) relative to 11 September 2019.

The action of strong winds on sea ice removal from the area at the mouth of the Amery Ice Shelf was visible in MODIS imagery (Fig. 8). During the period of the twin polar cyclones on 23- 24 September 2019, the sea ice was pushed about 65 km away from the ice shelf front in just a 2-day period of time (Fig. 8). The ice-free region in front of the ice shelf presented an asymmetric shape where the sea ice in front of the western side of the ice shelf was pushed further away compared to the sea ice in front of the eastern side (Fig. 8). This may have made the western side more vulnerable to the winds and associated waves induced by the consecutive explosive cyclones.

In fact, sea ice loss in the vicinity of weakened or flooded shelves is considered as the ultimate cause of rapid ice shelf calving (e.g., Massom et al., 2018). The removal of the protective buffer represented by sea ice for ice shelves (e.g., Massom et al., 2018) may have enabled increased flexure of the outer ice shelf western margin by wind-induced waves.

The extreme nature and long duration of the cyclones during the second event induced sustained wind strain on the pre-existing rift at the front of the Amery Ice Shelf, where winds were blowing perpendicular to it. This caused rapid opening of the crack and subsequent movement of the iceberg away from the ice shelf (Fig. 8). The SAR satellite image on 23 September 2019 together with the ice-displacement velocity (rate and direction in vectors) relative to 11 September 2019 are shown in Fig. 7g. The displacement vectors indicate that the iceberg-to-be was rotating in the period 11- 23 September 2019 prior to the calving. Wind forcing induced a leftward (relative to the rift T1) splitting-movement of the future iceberg prior to break-off on 25 September 2019.

Explosive cyclones crossing the sea ice zone around Antarctica can generate waves of up to 8 meters in height that are capable of propagating more than 100 km into the sea ice cover (Vichi et





al., 2019). The consecutive deep cyclones under scrutiny impacted immediately the Amery Ice
Shelf front since they were found very close to the coast. During the first period of explosive twin
cyclones, the cyclones were sitting to the west of the Amery Ice Shelf which directed anomalous
warm and moist easterlies towards it. This situation caused a decrease in sea ice concentration
(Fig. 7) and intense waves immediately at the shelf front (Fig. 4f) resulting in a cumulative shelf-
front fatigue and amplification of the fractures along the pre-existing rifts. This combination of
factors weakened the ice shelf front and made it more vulnerable to the additional extreme
atmospheric forcing (producing strong offshore winds) and sea ice removal brought by the second
event of explosive twin cyclones, ultimately leading to its calving.
Previous studies (Holdsworth and Glynn, 1978; Squire et al., 1994) have shown that calving ice
shelves can be triggered by wind-induced waves which impose flexural strains on the ice shelves,
with the potential to induce crevasse and rift propagation and calving (Robinson and Haskell, 1992;
Bromirski et al., 2010). This effect can be even maximized by the loss of the protective sea ice
pack at the front of the ice shelves (Massom et al., 2018). Here we have shown that the series of
intense cyclones provided ideal conditions for both sea ice reduction and wind-waves and
ultimately triggered the calving on 25 September 2019.
Furthermore, ocean swell (defined as relatively long-period surface-gravity waves that are
generated by distant weather systems and are no longer growing or being sustained locally by the
wind, as opposed to locally generated wind waves), may have also contributed to (i) fragilizing
the western side of the Amery Ice Shelf on 22 September 2019 (i.e., after the decay of the first two
twin cyclones) and to (ii) the calving on 25-26 September (i.e., after the decay of the second pair
of deep cyclones). Moreover, swells are strongly attenuated by the presence of extensive sea ice
which reduced substantially their destructive effect. Thus, loss of sea ice can maximize swell effect
on ice shelves. This mechanism has been found at work during the calving events of other Antarctic
ice shelves. A study on the calving event in March 1990 at the Erebus Glacier Tongue in the Ross
Sea implicated the removal of sea ice combined with ocean swell (Robinson and Haskell, 1990).
Focusing on the disintegration of the Larsen ice shelves, Massom et al., (2018) found that regional
loss of sea ice before and during the disintegration events allows storm-induced long-period (10–
20 s) ocean swells to reach exposed ice shelf fronts that have been preconditioned for calving by
extensive fracturing and meltwater flooding. These swells excite flexural oscillations in the outer
ice shelf margin which amplify fracture and trigger a calving.


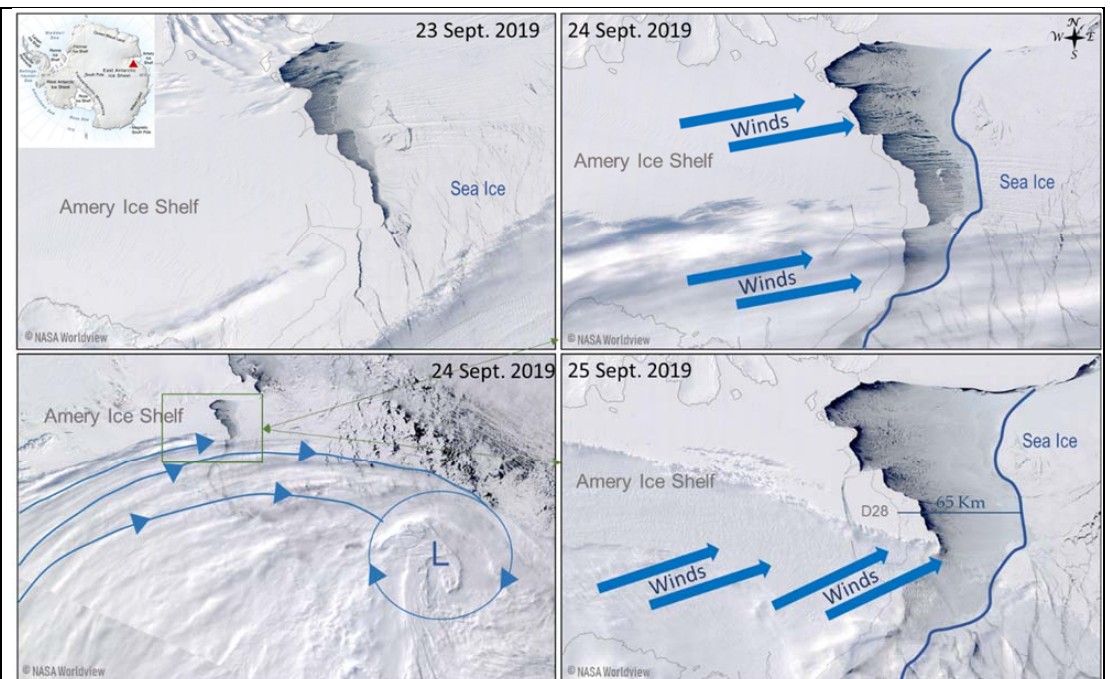

**Figure 8:** MODIS satellite visible imagery of the Amery Ice Shelf showing the ice shelf before the calving on 23 September 2019, during the calving on 24-25 September 2019. Image credit: NASA Worldview.


## 3. Discussion and conclusions


In this study, the role of atmospheric extremes in ice shelf instability is addressed by investigating
the atmospheric conditions during the recent calving event from the Amery Ice Shelf on 25
September 2019. During the month of September 2019, the circulation around Antarctica was
characterized by anomalously-pronounced 3 ridges and 3 troughs with the Indian sector of the
Southern Ocean being under the influence of troughing and surrounded by two blocking ridges;
one over the southern Atlantic to the west and one over Davis Sea and southern Australia to the
east.
During the second half of September 2019, a series of explosive polar cyclones, evolving into
stationary twin polar cyclones, impacted the region of the Amery Ice Shelf. The first explosive
cyclone occurred on 18 September 2019 and evolved into two stationary twin polar cyclones on
19-22 September 2019 sitting to the west of the ice shelf. The second explosive cyclone formed
on 23 September 2019 and evolved into two stationary twin polar cyclones on 24-25 September
2019, sitting, this time, to the east of the Amery Ice Shelf. Both explosive-cyclone episodes were
accompanied by intense atmospheric rivers bringing anomalous warm and moist air masses
poleward. The stationary aspect of the deep cyclones had a large impact on the ice conditions as it
subjected the ice to sustained stress and strain. The main difference between the two episodes is
the location at which the twin cyclones were stationary, relative to the Amery Ice Shelf, which



determined the characteristics of the air masses and the wind direction that affected the ice shelf.
This position of the cyclones relative to the Amery Ice Shelf was, in turn, determined in each
episode, by the location of the blocking ridges in the general circulation.
During the first episode, anomalous warm, moist and easterly winds impacted the ice shelf and
surrounding sea ice, whereas during the second episode, the ice shelf and surrounding sea ice were
under the influence of anomalous westerlies. The first episode resulted in a weakened and more
exposed ice shelf due to (i) sea ice melting, and (ii) fracture amplification along the rift via wind-
induced waves parallel to it. During the second episode, anomalously strong offshore winds
resulted in an ice-free area in front of the western side of the ice shelf. Sustained strong winds
perpendicular to the pre-existing rift induced (i) significant sea ice drift away from the shelf-front,
(ii) sustained strain on the shelf-front, and (iii) fracture amplification along the rift leading to the
calving. The detached iceberg after calving followed a northeasterly motion being dragged by the
prevailing winds and associated ocean currents. This drifting direction was similar to the one
followed by the sea ice one day before and gave an indication of the impact of the wind direction
on this process. Given the east-exposed orientation of the crack at the ice shelf-front, the direction
of the sustained strong westerly winds was deterministic for the calving.
In summary, atmospheric forcings by the explosive twin polar cyclones and associated
atmospheric rivers, triggered the September 2019 breakoff at the Amery Ice Shelf via the interplay
between several mechanisms:
- Thermodynamical processes linked to the poleward transport of anomalous warm and moist air
masses which resulted in a reduction in sea ice concentration at the front of the ice shelf, thus
increasing the exposure of the ice shelf-front to winds and waves.
- Mechanical forcing by the anomalously strong winds on: 1- Waves/swells which resulted in a
cumulative shelf-front fatigue and amplification of the fractures along the pre-existing rift. 2- Sea
ice drift which removed the protective buffer and increased the effect of wind-induced waves on
the ice shelf margin. 3- The rift which led to the enlargement of the fracture and separated the
iceberg from the shelf along the rift. 4- Subsurface warm waters which may have generated an
influx of warm circumpolar deep water onto the ice shelf margin and under the ice shelf cavity
that could have preconditioned the breakoff through basal melt (Morrison et al., 2020). The last
point has not been explored in this study since it focuses on the role of atmospheric forcing on the
calving.
The analysis of this unique event could help better understand the underlying factors leading to
calving and ice shelf weakening; important precursors to ice shelf instability and disintegration
and hence contribution to sea level rise. Our analysis highlights the need for ice sheet models, used
to project sea level rise, to account for atmospheric forcing at high resolution, in addition to sea
ice and ocean waves, if they were to simulate accurately the changes occurring in the ice sheet and
glaciers and their contribution to sea level rise. Up till now, the role of warming ocean has been
the focus of most of the scientific research on this topic, however, recent studies based on
numerical experiments demonstrate that wind stress changes over the Southern Ocean drive
enhanced poleward heat transport by stronger subpolar gyres and reduce coastal sea ice and cold-





water formations, both of which result in an increased ocean heat flux into Antarctic ice shelf cavities (Kusahara, 2020). Furthermore, the increase of sea ice–free days can lead to enhanced regional subwater contribution to the basal melting.

In fact, important changes in the atmospheric circulation are being observed in the Southern Hemisphere. For instance, between 1979 and 2010 the subtropical jet streams moved poleward by $6.5 \pm 0.2$ degrees in the Southern Hemisphere (Hudson, 2012) and the westerlies strengthened and shifted poleward (Fogt and Marshal, 2020). The observed poleward movement over the past few decades represents a significant change in the position of the sub-tropical jet stream, which should lead to significant latitudinal shifts in the global weather patterns, the hydrological cycle and their impact on Antarctic ice shelves.

The variability of the polar jet front in the Southern Hemisphere and whether similar behavior as the polar jet in the Northern Hemisphere is underway around Antarctica needs to be investigated in future work. Several studies have shown evidence for a wavier jet stream in response to rapid Arctic warming and reported a weakening of the polar jet as a result of a reduced temperature gradient between high and mid-latitudes due to the increased temperatures in the Arctic (e.g., Francis and Vavrus, 2015; Coumou et al., 2015; Mann et al., 2017). Such change in the polar jet, which acts as an isolation boundary between high and mid latitudes, would lead to more interactions, and spark feedback mechanisms between the Antarctic system and mid-latitudes as it happened to be the case in the Arctic (Francis et al., 2018; 2019a).

The poleward shift of the cyclones together with the decrease in sea ice extent in recent years makes it more urgent to assess the impact of cyclones on Antarctic-wide maritime-terminating ice shelves as higher numbers of large cyclones could be expected to reach further south and therefore affects ice shelves dynamics. If extreme polar cyclones are to form or reach more frequently ice shelves due to climate change, their destructive effect may have important consequences and needs to be accounted for in models used for sea level and Antarctic Ice Sheet mass balance projections.

## 4. Data and methods

The atmospheric analysis is based on data from the ERA5 reanalysis (Hersbach et al., 2020). During the period 16-25 September 2019, hourly maps of mean sea level pressure (MSLP), winds, 2m temperature and total column water vapor (TCWV) are analyzed. Furthermore, in order to investigate the anomalous character of the atmospheric conditions, we calculated, for the same period and quantities listed above, hourly standardized anomalies and percentile ranks relative to all hourly ERA5 September values during the full record (1979-2019) over the area 45-95°E, 50-75°S. In addition, a histogram analysis has been performed over a smaller domain limited to the ice-shelf front area and adjacent mouth of Prydz Bay (i.e., 70-65°S and 70-75°E). The histograms represent the distribution of hourly values spatially averaged over this domain, for all months during 1979-2019.

Daily sea ice extent and concentration data are derived from the AMSR-E / AMSR2 unified record (Meier et al., 2018) at 12.5 km spatial resolution (https://nsidc.org/data/AU_SI12/versions/1). To check the motion in the sea ice field in the Amery Basin, we used the low-resolution sea ice drift product of the EUMETSAT Ocean and Sea Ice Satellite Application Facility (OSI SAF, www.osi-



saf.org). This is a 48-hour average gridded ice drift dataset processed on a daily basis and made available on a 62.5 km Polar Stereographic Grid (e.g., Kwok et al. 2017). Ice motion vectors are estimated by an advanced cross-correlation method on pairs of satellite images (Lavergne et al., 2010). It uses the multi-sensor spatial covering product that combines SSMIS (91 GHz H and V polarization) on board DMSP platform F17, ASCAT (C-band backscatter) on board EUMETSAT platform Metop-A, and AMSR-2 on board JAXA platform GCOM-W. Due to atmospheric noise and surface melting these data are only available for the Southern Hemisphere winter (1st April to 31st October).

Visible imagery of the Amery Ice Shelf and surrounding area are taken from MODIS/VIIRS land products (ORNL DAAC, 2018) using the NASA Worldview application (https://worldview.earthdata.nasa.gov).

Sentinel-1 data has been used to determine potential surface melt (e.g. Datta et. al, 2019) and to track ice velocity over the Amery ice shelf prior to the D28 iceberg calving by using feature tracking in ESA's SNAP Sentinel-1 toolbox. Sentinel-3A and 3B data were used via the ESA Ocean Virtual Laboratory application to determine the wave height at the front of the Amery Ice Shelf.

## Acknowledgments

We acknowledge the use of imagery from the NASA Worldview application (https://worldview.earthdata.nasa.gov), part of the NASA Earth Observing System Data and Information System (EOSDIS). This work was supported by Masdar Abu Dhabi Future Energy Company, United Arab Emirates, Grant 8434000221. The contribution of Petra Heil was supported by the Australian Government's Australian Antarctic Partnership Program, and contributes to AAS Project 4506.

**Code and Data availability:** All data needed to evaluate the conclusions in the paper are present in the paper. Correspondence and requests for materials should be addressed to DF.

**Author contributions** D.F. conceived the study and wrote the initial manuscript. K.M. analyzed the satellite and reanalysis data. S.L analyzed satellite data. M.T. and P.H. provided input on result analysis. All authors interpreted results and provided input to the final manuscript.

**Competing Interests** The authors declare that they have no competing interests.



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
