# Peer review of "Atmospheric extremes triggered the biggest calving event in more than 50 years at the Amery Ice shelf in September 2019"

_The Cryosphere, 2020_

## Referee Comment (RC1) · Anonymous Referee #1 · 12 Sep 2020

This submission deals with the weather and synoptic conditions occurring at the time of the dramatic breakoff of iceberg D28 from the Amery Ice Shelf in September last year. The study explores, and implicates, the conditions of the forcings from twin polar explosive cyclones and their associated consequences. The analysis is comprehensive and considers the effects of the dynamics (very anomalous strong winds, stress, active wave field, and moisture transports) and moist thermodynamics (transport of sensible and latent heat) and other relevant issues.

The text and the logical structure of the paper make it easy to read, and it presents some important insights. I requesting that the authors revise the manuscript in line

with the comments I make below.

As part of the Introduction (and context) and perhaps later discussion it would be valuable to reference the recent analysis (and perspectives) of Teng Li, Yan Liu and Xiao Cheng, 2020: Recent and imminent calving events do little to impair Amery ice shelf's stability. Acta Oceanologica Sinica, 39, 168-170, doi: 10.1007/s13131-020-1600-6. Similar comment with respect to the earlier, but still very relevant, investigation of Simone Darji, Sandip R. Oza, R. D. Shah, B. P. Rathore and I. M. Bahuguna, 2018: Rift assessment and potential calving zone of Amery Ice Shelf, East Antarctica. Current Science, 115, 1799-1804, doi: 10.18520/cs/v115/i9/1799-1804.

LINE 97 – 101 – In this context very beneficial to also reference the studies of Rudeva et al., 2015: Variability and trends of global atmospheric frontal activity and links with large-scale modes of variability. J. Clim, 28, 3311-3330, and Pezza and co-authors, 2007: Southern Hemisphere cyclones and anticyclones: Recent trends and links with decadal variability in the Pacific Ocean. Int. J. Climat., 27, 1403-1419.

LINES 101-103 - Questions have been raised in the literature in connection with the interpretation of 'cyclone' statistics and their trends. There are many automated cyclone identification schemes and the different choices made in these can give rise to different results (refer here to, e.g., Neu, U. et al., 2013: IMILAST: A community effort to intercompare extratropical cyclone detection and tracking algorithms. Bull. Amer. Meteor. Soc., 94, 529-547). Regarding the potential influence of this on the analysis of future trends very helpful here to cite the investigation of U. Ulbrich, Gregor C. Leckebusch, Jens Grieger, et al., 2013: Are greenhouse gas signals of Northern Hemisphere winter extra-tropical cyclone activity dependent on the identification and tracking algorithm? Meteor. Zeitschrift, 22, 61-68 who find robustness across state-of-the-art cyclone schemes for the NH.

LINE 106 – Note that Jim Kossin's paper deals with tropical cyclones, rather than extratropical cyclones. In the context here (of ETCs), including this reference is perhaps

misleading.

LINE 107-119 As part of this broad Introduction it would be helpful to refer to the analysis of Uotila, Vihma, et al., 2011: Relationships between Antarctic cyclones and surface conditions as derived from high resolution NWP data. J. Geophys. Res., 116, doi: 10.1029/2010JD015358 for quantification of the many interactions between Antarctic cyclones and surface parameters of relevance to the present study.

LINES 112-114 – As made clear here, these cyclones and 'rivers' transport poleward significant amounts of heat and moisture. Recent investigations have revealed the significant consequences in the polar regions of increased downward longwave radiation on sea ice melt and temperature. This important radiative aspect should be explicitly mentioned, and reference made to Lee, Gong, et al., 2017. 'Revisiting the cause of the 1989-2009 Arctic surface warming using the surface energy budget: Downward infrared radiation dominates the surface fluxes', Geophys. Res. Lett. 44, 10,654–10,661.

LINE 113 – Here (and in a number of other places in the text) the authors refer to Francis et al. (2019). However, in the References there are details of Francis et al. (2019a) and Francis et al. (2019b) (lines 594-599). Please sort this out.

LINES 115-119 - There is a significant amount of research and new insights now on this key topic. Consider updating the references with Vernon A. Squire, 2020: Ocean wave interactions with sea ice: A reappraisal. Annual Review of Fluid Mechanics, 52, 37-60, doi: 10.1146/annurev-fluid-010719-060301. Vernon A. Squire, 2018: A fresh look at how ocean waves and sea ice interact. Philosophical Transactions of the Royal Society A - Mathematical Physical and Engineering Sciences, 376, 20170342, doi: 10.1098/rsta.2017.0342.

LINES 192-202 – The authors speak here of the synoptic conditions exhibiting an amplified zonal wave number 3 (ZW3). I strongly suggest wording this more carefully. 'zonal wave number 3' can broadly speaking be thought of as three ridges (or troughs) spaced approximately equally around the SH. However, it is more rigorously taken to

be the quantitative metric defined by Marilyn Raphael (2004, 2007) as the average normalised geopotential (or pressure) at the three points (49S, 50E), (49S, 166E), and (49S, 76W). A problem that arises from this fixed-locations definition is that the 3-wave structure shows considerable longitudinal variability. In fact, the 3-wave structure shown in Fig. 2a is shifted to the west of the above points by some 35 degrees, i.e., over a quarter of a wavelength. Recent research has been directed at defining a ZW3 which allows for these translations (make reference here to Irving and co-authors (2015) A novel approach to diagnosing Southern Hemisphere planetary wave activity and its influence on regional climate variability. J. Clim. 28, 9041-9057). To avoid misleading the reader on this important point please make some more targeted comments here (and where relevant elsewhere) in the paper.

LINE 216-217 – To avoid any NH/SH confusion I suggest changing ' . . . to the left of the low-pressure center' to ' . . . to the west of the low-pressure center'. While 'left' is strictly correct for this limited region plot, it has the potential to obscure the physical relationship. Please make similar changes to 'left' at lines 219, 342, . . . .

LINE 225 – You have used 'standard deviations' up till now. Perhaps be consistent and replace this for 'sigma'. (Similar comment for line 242.)

LINE 277-296 – Figure 5 shows lots of very interesting structure. Two panels show IVT and TCWV, but it would be interesting to also show precipitation. While I don't absolutely require this, it would be valuable in indicating the magnitude of the lower-atmosphere latent heat release and its potential role in driving these extreme systems.

LINE 318-319 – Figure 6f presents much valuable information in a neat form.

LINE 623-624 – Please present full details. They are . . . Hersbach, H., B. Bell, P. Berrisford, et al., 2020: The ERA5 global reanalysis. Quarterly Journal of the Royal Meteorological Society, 146, 1999–2049, doi: 10.1002/qj.3803.

LINE 766-768 – Wille, J. D., V. Favier, A. Dufour, I. V. Gorodetskaya, J. Turner, C.

Agosta and F. Codron, 2019: West Antarctic surface melt triggered by atmospheric rivers. Nature Geoscience, 12, 911-916, doi: 10.1038/s41561-019-0460-1.

---

## Referee Comment (RC2) · Ted Scambos (Referee) · 30 Sep 2020

While I've given low marks on the paper as it stands, I think this assemblage of weather and sea ice data can still be applicable if the interpretation of events and causality is changed.

The study discusses the events surrounding the calving of the D28 iceberg (US National Ice Center designation). Two intense polar lows, each resulting in twinned cyclones (the latter one resulting in a pair of twinned cyclones) that brought both intense winds, significant local wave action, high moisture and warm air advection from midlatitudes, and significant sea ice motion. The authors interpret the combined effects of

wind-driven waves, and wind stress itself, to lead to the calving event of 25 September 2019.

There are several mis-steps in this interpretation, although the weather data that has been marshalled and the analysis in terms of statistics and the hour-by-hour unfolding of the events is useful and impressive. In short, however, it is unlikely that wind stress and local wave action can be directly linked to the calving. I urge the authors to look instead at the effect of sea surface slope, both due to tides and to wind-driven ocean movement – storm surge and its opposite – an offshore oceanward slope that leads to gravitationally-driven calving along the existing rift. These are the forces that drive short-term movement of icebergs in the open ocean (resulting in their spiral or cycloidal motion – a result of tides coming and going, i.e., changes in ocean surface slope).

It is unclear to me how the first event preconditions the ice shelf to calving, since the two major impacts of the intense storm of Sept 18-19, 2019 are onshore and easterly winds, and an intense snowfall on the ice shelf (strong moisture advection, 'warm' air for the season, but still sub-freezing conditions). Wave action by long-period gravity or wind-driven waves is unlikely because of the damping effect of sea ice – the only open ocean areas are small and near the coast, which would not provide sufficient fetch to build multi-meter wave heights or long-period gravitational waves.

With regard to wind stress: while intense, I would expect that the observed, mostly offshore, winds at the ice front area for the second event (23-24 September) are -not- the strongest winds the Amery Ice Shelf front has seen in the past nearly 60 years, or even in the past decade when the ice front and rifts were more or less the same as they are now. Winds in excess of 35 m/s, i.e., far stronger than the winds at the ice front during these events, are not particularly rare around Antarctica's ice edge or coast in my experience (at McMurdo, the Peninsula, the Amundsen Sea coast, etc.). Being in the 99th percentile means that, on average, winds at that level or higher occur ∼85 hours out of every year. The Amery front will have seen considerably stronger winds than the 23-24 September event at some point in the past (say) 10 years.

With regard to wave action: to have a significant impact on an ice shelf, the wavelength of the incoming wave must be an appreciable fraction of the ice thickness – tens to hundreds of meters (10s of seconds in period) – and that is rare, and part of the reason why ice fronts on ice shelves survive for decades without fragmenting. Moreover, while long-period waves can penetrate an open pack much farther than short-period waves ('chop' or 'swell' of a few meters wavelength), even the longest wavelength waves of a wind-driven nature do not retain any significant amplitude after 50 km inside the pack – and in this case the Amery was shielded from the longest-wavelength, highest amplitude, open ocean waves by over 500 km of ice pack. Note that in autumn, every year, the ocean area in front of the Amery is generally ice-free, and waves from the Southern Ocean impact the ice shelf front without damping.

However – at still longer periods and wavelengths, beyond gravity waves or the longest-period ocean waves, there are tides, inverse barometer effects, and storm-driven surges, both towards and away from the ice front. The pattern of storms described, and the wind field of 23-24 September, implies that there must have been a significant slope on the ocean surface at/near the ice front; and between the previous storm and the 23-24 September storm there would likely be a rapid change from a shoreward surge to ocean slope away from the ice front. This oceanward slope tugs on the ice front, placing the extensional stress you note in Figure 7g on the rift, leading to rift growth and calving.

While my suggestions are not checked out for this event, have a look at these references. Please consider tides and wind-driven ocean slope, and consider if that process makes a more likely case for this calving.

Bennetts, L.G. and Squire, V.A., 2012. On the calculation of an attenuation coefficient for transects of ice-covered ocean. Proceedings of the Royal Society A: Mathematical, Physical and Engineering Sciences, 468(2137), pp.136-162.

Brunt, K.M., Okal, E.A. and MacAYEAL, D.R., 2011. Antarctic ice-shelf calving triggered by the Honshu (Japan) earthquake and tsunami, March 2011. Journal of Glaciology, 57(205), pp.785-788.

MacAyeal, D.R., Okal, M.H., Thom, J.E., Brunt, K.M., Kim, Y.J. and Bliss, A.K., 2008. Tabular iceberg collisions within the coastal regime. Journal of Glaciology, 54(185), pp.371-386.

Scambos, T., Ross, R., Bauer, R., Yermolin, Y., Skvarca, P., Long, D., Bohlander, J. and Haran, T., 2008. Calving and ice-shelf break-up processes investigated by proxy: Antarctic tabular iceberg evolution during northward drift. Journal of Glaciology, 54(187), pp.579-591.

Squire, V.A., 2020. Ocean wave interactions with sea ice: a reappraisal. Annual Review of Fluid Mechanics, 52, pp.37-60.

Detailed comments follow –

To summarize my main issue with the first few pages of this manuscript (and to paraphrase the Bible): Render unto climate change that which is climate change; Render unto calving processes that which belongs there. ïĄŁ

Line 18 – ..play 'a' crucial role. . .

Abstract could be significantly shorter and more quantitative, terser and a bit less dramatic (unpredictable, unexpected, explosive, record-anomalous). It almost has the feel of a press release – press releases are not bad, but they're not peer-review abstracts.

Keywords: 'Twin polar cyclones' is not a keyword that anyone is going to use to find your paper. Same with 'explosive cyclones' I'll bet, but perhaps it's worth noting. Also, the Amery is not unstable, and stable ice shelves have to endure weather extremes (and, in some cases, they calve large bergs as a result). How about: Ice shelf calving, icebergs, Amery Ice Shelf, Antarctica, blocking highs, polar cyclones, explosive cyclones.

Line 42 – I would remove 'the': '. . .response to globally rising temperatures and to. . .'

[Figure]

Line 46 – change to '. . .key catchment basin in East Antarctica. . .'. Also, remove 'con­sidered' – it is measured as being in balance, that's not an opinion.

Line 57 – change to '. . .would not experience a major calving until around 2025 or later.' (these estimates by Fricker and the other papers are very approximate)

Lines 60-61 – the last half of the sentence. Also, the following paragraph, Lines 62-80. This is a bit off, for the reader there's going to be confusion or mis-understanding. Ice shelves are not 'weakened' by this kind of atmospheric forcing, and the stability of the shelf as a whole does not change very much. Your study is about the link between an extreme weather event and a rapid increase in rift propagation followed by calving of a large iceberg. But iceberg calving is a normal process of stable ice shelves. Some event immediately precedes the calving in all cases – in this case, a major storm. But this process and result are not of the same genre as climate-induced, esp. melt-driven, changes in ice shelf stability: it is a study of how calving can be triggered by an extreme wind event.

Line 68 – Not In This Case – the calving of the front of the Amery has almost NO IMPACT on the resistance or longitudinal backstress on the Lambert Glacier – See Furst et al, 2016, or any study of the strain rates on ice shelves and the adjacent grounded ice. Once past the lateral shear zones, or if the shear zones are disrupted to the point of having no significant mechanical contact, there is no backstress provided by the shelf ice. . ..

Line 72-73 – -if- you can figure out how to keep this review discussion and still relate it to your calving study, cite Scambos et al., 2014 here - -broader in scope, more general for the northern Peninsula.

Line 77 – Aitken is talking about a completely different area, with -increasing- basal melting, not steady-state basal melting a the grounding line that most ice streams/ice shelves experience at their grounding zones. . .

Your study is about an unusual large calving event – it's not about climate change or ice shelf stability in the lines of Pritchard et al, or Furst et al, or the several papers by Khazendhar... or Aitken, or Dutrieux, or Scambos.. or several by Rignot.. .its a calving event. You have a nice link to a weather anomaly. But that is as far as it should be taken.

There -is- a need to understand calving processes and the 'end-game' controls on release of an iceberg, and the most important reason is modelling studies of future evolution. There is a rich body of work on this need. Stick to the justifications in the Fricker, Bassis, etc. papers for this paper.

Line 81 – 92 – although this paragraph also needs extensive re-writing, in your review of the links between physical phenomena and all kinds of calving, please also cite Brunt et al., 2011, https://doi.org/10.3189/002214311798043681

Line 93-96, 97-106, 107-onward – this is more appropriate, introductory, justification-for-study text. Your introduction is too long anyway – a more cursory discussion of climate-change-related ice shelf and glacier impacts is needed, briefly, and make it clearer that it is only obliquely related to the results of this study. Doing this will make your paper tighter.

Line 115-116 – Re-state: wave amplitude is reduced by several orders of magnitude within 10km of the sea ice edge; yes, cm-scale, long period (30 sec or longer) waves can penetrate the pack for long distances. Make that clearer.

Figure 1... One of the image maps needs a scale bar. Check north arrow, also. I find that a few lat-long lines are better for Antarctica anyway, where the direction of north can vary from one side of the image or map to the other.

Line 208 – '….then decayed on 19 September at 1300 UTC.' Suggest you re-write this with something like, '…then decayed on 19 September, with 9XX hPa central pressure by 1300 UTC.'

Line 220-230 – might want to note that the atmospheric river (need to introduce the acronym before using it), in carrying a lot of moisture toward the ice sheet and upward, was inevitably warmed by condensation (released heat) as well as by downward flow, a kind of foehn event.

Seems like you could eliminate a few of the 8 panels in Figures 2 and 3. Not sure you need to show the anomalies (or perhaps move them to supplemental information) Line 242 – do you have an estimate of the wind speed at an Amery AWS during this period?

Fig 4g – won't this just be a measure of the sea ice ridging in the region in the aftermath of the intense north-northeasterly first cyclone and first twin cyclone?

Line 251 – 'meandered' is an informal term, and means a sinuous path – is that the case?

Line 281 –change to ' far-south-reaching blocking highs'.

Line 290 – change to 'advected very cold air (2 m temperatures of ..'

Line 293-296 – these sentences have redundant information, shorten: the atmospheric river continued to advect large amounts of moisture and very warm air to the ice shelf.

Line 301-317 and elsewhere – The authors frequently use percentiles and standard deviations in the text to describe the extreme conditions, but these are difficult to appreciate without a lot of background awareness of typical weather in the region. While it does convey the unusualness of the conditions, I suggest shifting the description here and elsewhere to -always- using the absolute values and bearings first (rather than u-values and percentiles, or std. dev. numbers) and then put the statistically-based assessments in parentheses after the 'real' numbers. Readers can more easily grasp '25 m/s winds' relative to '99 percentile (5 std dev above mean) winds'.

Line 335-338 – I don't see how this would be the case. Offshore winds compact and compress the frazil ice into thicker ridged ice, and expose more ocean surface, a kind of sea ice factory. This compressed ice cools and becomes part of the pack, and is

generally quite durable because of the wind compression.

Line 339-345 – I really doubt this – you are showing more than 500 km of mostly 100% pack ice – even a few km of pack ice dramatically reduces the effect of swell! This line of argument must be removed. The small open areas you show in Figure 7 -might-develop a bit of chop and swell -in the fetch of the <50% ice cover areas alone- and that's about it.

Line 346-351 – No again! If there was an intense moisture plume in the events (the IVT 'river') then there was intense snowfall over the sea ice, thickening it and toughening it. The new areas of low concentration you show are -local- and due to intense wind compaction in the leeward direction. And, Lines 349-351, the local waves induce insignificant flexure or vibration of the ice shelf, almost zero, the ice is far too thick, any flexure would be completely elastic.

Lines 358-368 – authors are on the wrong track. Much of this and the preceding ∼100 lines are speculation.

Lines 369-409 – mostly speculation, and wrong direction. . ... The kind of high-frequency swell generated by high winds in closed near-shore polynyas would have a trivial to non-existent effect on a shelf hundreds of meters thick, cold-ice-cored, like the Amery.

Ok, I stopped here. This is off track. See opening discussion of the work.

---

## Author Comment (AC1) · 13 Nov 2020

**Manuscript TC-2020-219:**

**Atmospheric extremes triggered the biggest calving event in more than 50 years at the Amery Ice shelf in September 2019.**

*By D. Francis et al.*

**Reply to reviewers' comments**

The authors would like to thank the reviewers for their valuable comments which will help to improve the quality of the manuscript. Kindly find below in blue our response point-by-point to the reviewers' comments and suggestions.

**Reviewer #1:**

This submission deals with the weather and synoptic conditions occurring at the time of the dramatic breakoff of iceberg D28 from the Amery Ice Shelf in September last year. The study explores, and implicates, the conditions of the forcings from twin polar explosive cyclones and their associated consequences. The analysis is comprehensive and considers the effects of the dynamics (very anomalous strong winds, stress, active wave field, and moisture transports) and moist thermodynamics (transport of sensible and latent heat) and other relevant issues. The text and the logical structure of the paper make it easy to read, and it presents some important insights. I requesting that the authors revise the manuscript in line with the comments I make below.

Thank you for your positive feedback on the merit of our manuscript. Our response to the specific comments is below.

1) As part of the Introduction (and context) and perhaps later discussion it would be valuable to reference the recent analysis (and perspectives) of Teng Li, Yan Liu and Xiao Cheng, 2020: Recent and imminent calving events do little to impair Amery ice shelf's stability. Acta Oceanologica Sinica, 39, 168-170, doi: 10.1007/s13131-020-1600-6.

Agreed.

2) Similar comment with respect to the earlier, but still very relevant, investigation of Simone Darji, Sandip R. Oza, R. D. Shah, B. P. Rathore and I. M. Bahuguna, 2018: Rift assessment and potential calving zone of Amery Ice Shelf, East Antarctica. Current Science, 115, 1799-1804, doi: 10.18520/cs/v115/i9/1799-1804.

Agreed.

3) LINE 97 – 101 – In this context very beneficial to also reference the studies of Rudeva et al., 2015: Variability and trends of global atmospheric frontal activity and links with large-scale modes of variability. J. Clim, 28, 3311-3330, and Pezza and co-authors, 2007: Southern Hemisphere cyclones and anticyclones: Recent trends and links with decadal variability in the Pacific Ocean. Int. J. Climat., 27, 1403-1419.

Agreed.

4) LINES 101-103 - Questions have been raised in the literature in connection with the interpretation of 'cyclone' statistics and their trends. There are many automated cyclone identification schemes and the different choices made in these can give rise to different results (refer here to, e.g., Neu, U. et al., 2013: IMILAST: A community effort to intercompare extratropical cyclone detection and tracking algorithms. Bull. Amer. Meteor. Soc., 94, 529-547). Regarding the potential influence of this on the analysis of future trends very helpful here to cite the investigation of U. Ulbrich, Gregor C. Leckebusch, Jens Grieger, et al., 2013: Are greenhouse gas signals of Northern Hemisphere winter extra-tropical cyclone activity dependent on the identification and tracking algorithm? Meteor. Zeitschrift, 22, 61-68 who find robustness across state-of the-art cyclone schemes for the NH.

Agreed.

5) LINE 106 – Note that Jim Kossin's paper deals with tropical cyclones, rather than extratropical cyclones. In the context here (of ETCs), including this reference is perhaps misleading.

Agreed. We will replace this reference by a reference relevant to ETCs.

6) LINE 107-119 As part of this broad Introduction it would be helpful to refer to the analysis of Uotila, Vihma, et al., 2011: Relationships between Antarctic cyclones and surface conditions as derived from high resolution NWP data. J. Geophys. Res., 116, doi: 10.1029/2010JD015358 for quantification of the many interactions between Antarctic cyclones and surface parameters of relevance to the present study.

Agreed.

7) LINES 112-114 – As made clear here, these cyclones and 'rivers' transport poleward significant amounts of heat and moisture. Recent investigations have revealed the significant consequences in the polar regions of increased downward longwave radiation on sea ice melt and temperature. This important radiative aspect should be explicitly mentioned, and reference made to Lee, Gong, et al., 2017. 'Revisiting the cause of the 1989-2009 Arctic surface warming using the surface energy budget: Downward infrared radiation dominates the surface fluxes', Geophys. Res. Lett. 44,10,654–10,661.

Agreed.

8) LINE 113 – Here (and in a number of other places in the text) the authors refer to Francis et al. (2019). However, in the References there are details of Francis et al. (2019a) and Francis et al. (2019b) (lines 594-599). Please sort this out.

Done.

9) LINES 115-119 - There is a significant amount of research and new insights now on this key topic. Consider updating the references with Vernon A. Squire, 2020: Ocean wave interactions with sea ice: A reappraisal. Annual Review of Fluid Mechanics, 52, 37-60, doi: 10.1146/annurev-fluid-010719-060301. Vernon A. Squire, 2018: A fresh look at how ocean waves and sea ice

interact. Philosophical Transactions of the Royal Society A - Mathematical Physical and Engineering Sciences, 376, 20170342, doi: 10.1098/rsta.2017.0342.

Agreed. Thanks for pointing us to these references.

10) LINES 192-202 – The authors speak here of the synoptic conditions exhibiting an amplified zonal wave number 3 (ZW3). I strongly suggest wording this more carefully. 'zonal wave number 3' can broadly speaking be thought of as three ridges (or troughs) spaced approximately equally around the SH. However, it is more rigorously taken to be the quantitative metric defined by Marilyn Raphael (2004, 2007) as the average normalised geopotential (or pressure) at the three points (49S, 50E), (49S, 166E), and (49S, 76W). A problem that arises from this fixed-locations definition is that the 3-wave structure shows considerable longitudinal variability. In fact, the 3-wave structure shown in Fig. 2a is shifted to the west of the above points by some 35 degrees, i.e., over a quarter of a wavelength. Recent research has been directed at defining a ZW3 which allows for these translations (make reference here to Irving and co-authors (2015) A novel approach to diagnosing Southern Hemisphere planetary wave activity and its influence on regional climate variability. J. Clim. 28, 9041-9057). To avoid misleading the reader on this important point please make some more targeted comments here (and where relevant elsewhere) in the paper.

Thanks very much for this insight. Agreed, indeed the reference you are suggesting is more relevant and will be used.

11) LINE 216-217 – To avoid any NH/SH confusion I suggest changing ' ... to the left of the low-pressure center' to ' ... to the west of the low-pressure center'. While 'left' is strictly correct for this limited region plot, it has the potential to obscure the physical relationship. Please make similar changes to 'left' at lines 219, 342, ....

Agreed. The necessary changes will be made.

12) LINE 225 – You have used 'standard deviations' up till now. Perhaps be consistent and replace this for 'sigma'. (Similar comment for line 242.)

Agreed.

13) LINE 277-296 – Figure 5 shows lots of very interesting structure. Two panels show IVT and TCWV, but it would be interesting to also show precipitation. While I don't absolutely require this, it would be valuable in indicating the magnitude of the lower atmosphere latent heat release and its potential role in driving these extreme systems.

Agreed. We have added precipitation to figure 5 and will be included in the revised version of the manuscript.

14) LINE 318-319 – Figure 6f presents much valuable information in a neat form.

Thanks.

15) LINE 623-624 – Please present full details. They are ... Hersbach, H., B. Bell, P. Berrisford, et al., 2020: The ERA5 global reanalysis. Quarterly Journal of the Royal Meteorological Society, 146, 1999–2049, doi: 10.1002/qj.3803.

Done.

16) LINE 766-768 – Wille, J. D., V. Favier, A. Dufour, I. V. Gorodetskaya, J. Turner, C. Agosta and F. Codron, 2019: West Antarctic surface melt triggered by atmospheric rivers. Nature Geoscience, 12, 911-916, doi: 10.1038/s41561-019-0460-1.

Done.

---

## Author Comment (AC2) · 13 Nov 2020

**Manuscript TC-2020-219:**

**Atmospheric extremes triggered the biggest calving event in more than 50 years at the Amery Ice shelf in September 2019.**

*By D. Francis et al.*

**Reply to reviewers' comments**

The authors would like to thank the reviewers for their valuable comments which will help to improve the quality of the manuscript. Kindly find below in blue our response point-by-point to the reviewers' comments and suggestions.

**1) Reviewer #2:**

While I've given low marks on the paper as it stands, I think this assemblage of weather and sea ice data can still be applicable if the interpretation of events and causality is changed. The study discusses the events surrounding the calving of the D28 iceberg (US National Ice Center designation). Two intense polar lows, each resulting in twinned cyclones (the latter one resulting in a pair of twinned cyclones) that brought both intense winds, significant local wave action, high moisture and warm air advection from midlatitude, and significant sea ice motion. The authors interpret the combined effects of wind-driven waves, and wind stress itself, to lead to the calving event of 25 September 2019. There are several mis-steps in this interpretation, although the weather data that has been marshalled and the analysis in terms of statistics and the hour-by-hour unfolding of the events is useful and impressive.

We thank the reviewer for his insightful comments and suggestions that certainly will help to improve the quality of the manuscript and make its findings more robust.

In short, however, it is unlikely that wind stress and local wave action can be directly linked to the calving. **I urge the authors to look instead at the effect of sea surface slope, both due to tides and to wind-driven ocean movement – storm surge and its opposite – an offshore ocean ward slope that leads to gravitationally-driven calving along the existing rift.** These are the forces that drive short-term movement of icebergs in the open ocean (resulting in their spiral or cycloidal motion – a result of tides coming and going, i.e., changes in ocean surface slope). It is unclear to me how the first event preconditions the ice shelf to calving, since the two major impacts of the intense storm of Sept 18-19, 2019 are onshore and easterly winds, and an intense snowfall on the ice shelf (strong moisture advection, 'warm' air for the season, but still sub-freezing conditions). Wave action by long-period gravity or wind-driven waves is unlikely because of the damping effect of sea ice – the only open ocean areas are small and near the coast, which would not provide sufficient fetch to build multi-meter wave heights or long-period gravitational waves.

We have looked at the processes suggested by the reviewer and in fact found an offshore ocean ward slope preceding immediately the calving. This and the associated discussion will be added to the revised manuscript. Additionally, we are investigating this aspect using a wave sea ice model and results will be added to the revised manuscript if found relevant.

[Figure]

Ocean slope derived from HYCOM for an area close to the ice shelf front (green polygon).

- With regard to wind stress: while intense, I would expect that the observed, mostly offshore, winds at the ice front area for the second event (23-24 September) are -not the strongest winds the Amery Ice Shelf front has seen in the past nearly 60 years, or even in the past decade when the ice front and rifts were more or less the same as they are now. Winds in excess of 35 m/s, i.e., far stronger than the winds at the ice front during these events, are not particularly rare around Antarctica's ice edge or coast in my experience (at McMurdo, the Peninsula, the Amundsen Sea coast, etc.). Being in the 99th percentile means that, on average, winds at that level or higher occur ~85 hours out of every year. The Amery front will have seen considerably stronger winds than the 23-24 September event at some point in the past (say) 10 years.

Noted. This will be revised accordingly.

- With regard to wave action: to have a significant impact on an ice shelf, the wavelength of the incoming wave must be an appreciable fraction of the ice thickness – tens to hundreds of meters (10s of seconds in period) – and that is rare, and part of the reason why ice fronts on ice shelves survive for decades without fragmenting. Moreover, while long-period waves can penetrate an open pack much farther than short-period waves ('chop' or 'swell' of a few meters' wavelength), even the longest wavelength waves of a wind-driven nature do not retain any significant amplitude after 50 km inside the pack – and in this case the Amery was shielded from the longest-wavelength, highest amplitude, open ocean waves by over 500 km of ice pack. Note that in autumn, every year, the ocean area in front of the Amery is generally ice-free, and waves from the Southern Ocean impact the ice shelf front without damping. However–at still longer periods and wavelengths, beyond gravity waves or the longest period ocean waves, there are tides, inverse barometer effects, and storm-driven surges, both towards and away from the ice front.

Noted. Thanks for this valuable information. The discussion in the revised manuscript will be corrected to take into account these points.

The pattern of storms described, and the wind field of 23-24 September, implies that there must have been a significant slope on the ocean surface at/near the ice front; and between the previous

storm and the 23-24 September storm there would likely be a rapid change from a shoreward surge to ocean slope away from the ice front. This ocean ward slope tugs on the ice front, placing the extensional stress you note in Figure 7g on the rift, leading to rift growth and calving. While my suggestions are not checked out for this event, have a look at these references. Please consider tides and wind-driven ocean slope, and consider if that process makes a more likely case for this calving.

Indeed, looking at the sea surface slope variation with time at the area in front of the ice shelf, we were able to see the consequences of the processes you are describing. The new figure of surface slope will be added to the revised manuscript and the associated discussion.

Bennetts, L.G. and Squire, V.A., 2012. On the calculation of an attenuation coefficient for transects of ice-covered ocean. Proceedings of the Royal Society A: Mathematical, Physical and Engineering Sciences, 468(2137), pp.136-162.

Brunt, K.M., Okal, E.A. and MacAYEAL, D.R., 2011. Antarctic ice-shelf calving triggered by the Honshu(Japan)earthquake and tsunami, March2011. Journal of Glaciology, 57(205), pp.785-788.

MacAyeal, D.R., Okal, M.H., Thom, J.E., Brunt, K.M., Kim, Y.J. and Bliss, A.K., 2008. Tabular iceberg collisions within the coastal regime. Journal of Glaciology, 54(185), pp.371-386.

Scambos, T., Ross, R., Bauer, R., Yermolin, Y., Skvarca, P., Long, D., Bohlander, J. and Haran, T., 2008. Calving and ice-shelf break-up processes investigated by proxy: Antarctic tabular iceberg evolution during northward drift. Journal of Glaciology, 54(187), pp.579-591.

Squire, V.A., 2020. Ocean wave interactions with sea ice: a reappraisal. Annual Review of Fluid Mechanics, 52, pp.37-60.

These references will be cited in the revised manuscript.

Detailed comments follow:

1) To summarize my main issue with the first few pages of this manuscript (and to paraphrase the Bible): Render unto climate change that which is climate change; Render unto calving processes that which belongs there.

Noted. This will be revised accordingly.

2) Line 18 – ..play 'a' crucial role... Abstract could be significantly shorter and more quantitative, terser and a bit less dramatic (unpredictable, unexpected, explosive, record-anomalous). It almost has the feel of a press release – press releases are not bad, but they're not peer-review abstracts.

Noted. We will revise accordingly.

3) Keywords: 'Twin polar cyclones' is not a key word that anyone is going to use to find your paper. Same with 'explosive cyclones' I'll bet, but perhaps it's worth noting. Also, the Amery is not unstable, and stable ice shelves have to endure weather extremes (and, in some cases, they calve large bergs as a result). How about: Ice shelf calving, icebergs, Amery Ice Shelf, Antarctica, blocking highs, polar cyclones, explosive cyclones.

Nice suggestion, we will modify accordingly.

4) Line 42 – I would remove 'the': '...response to globally rising temperatures and to...'

Ok.

5) Line 46 – change to '...key catchment basin in East Antarctica...'. Also, remove 'considered' – it is measured as being in balance, that's not an opinion.

Ok.

6) Line 57 – change to '...would not experience a major calving until around 2025 or later.' (these estimates by Fricker and the other papers are very approximate)

Ok.

7) Lines 60-61 – the last half of the sentence. Also, the following paragraph, Lines 62-80. This is a bit off, for the reader there's going to be confusion or mis-understanding. Ice shelves are not 'weakened' by this kind of atmospheric forcing, and the stability of the shelf as a whole does not change very much. Your study is about the link between an extreme weather event and a rapid increase in rift propagation followed by calving of a large iceberg. But iceberg calving is a normal process of stable ice shelves. Some event immediately precedes the calving in all cases – in this case, a major storm. But this process and result are not of the same genre as climate-induced, esp. melt-driven, changes in ice shelf stability: it is a study of how calving can be triggered by an extreme wind event.

Noted. This will be modified to include these comments and avoid any ambiguity for the readers.

8) Line 68 – Not in This Case – the calving of the front of the Amery has almost NO IMPACT on the resistance or longitudinal backstress on the Lambert Glacier – See Furst et al, 2016, or any study of the strain rates on ice shelves and the adjacent grounded ice. Once past the lateral shear zones, or if the shear zones are disrupted to the point of having no significant mechanical contact, there is no backstress provided by the shelf ice....

Noted. We will remove this statement.

9) Line 72-73 – -if- you can figure out how to keep this review discussion and still relate it to your calving study, cite Scambos et al., 2014 here - -broader in scope, more general for the northern Peninsula.

Noted. The reference will be included.

10) Line 77 – Aitken is talking about a completely different area, with -increasing- basal melting, not steady-state basal melting at the grounding line that most ice streams/ice shelves experience at their grounding zones...Your study is about an unusual large calving event – it's not about climate change or ice shelf stability in the lines of Pritchard et al, or Furst et al, or the several papers by Khazendhar... or Aitken, or Dutrieux, or Scambos.. or several by Rignot.. .its a calving event. You have a nice link to a weather anomaly. But that is as far as it should be taken. There -is- a need to understand calving processes and the 'end-game' controls on release of an iceberg, and the most

important reason is modelling studies of future evolution. There is a rich body of work on this need. Stick to the justifications in the Fricker, Bassis, etc. papers for this paper.

Noted. The discussion will be revised to reflect these remarks.

11) Line 81 – 92 – although this paragraph also needs extensive re-writing, in your review of the links between physical phenomena and all kinds of calving, please also cite Brunt et al., 2011, https://doi.org/10.3189/002214311798043681

Noted.

12) Line 93-96, 97-106, 107-onward – this is more appropriate, introductory, justification for-study text. Your introduction is too long anyway – a more cursory discussion of climate-change-related ice shelf and glacier impacts is needed, briefly, and make it clearer that it is only obliquely related to the results of this study. Doing this will make your paper tighter.

Agreed. Will take this into account and restructure the introduction.

13) Line 115-116 – Re-state: wave amplitude is reduced by several orders of magnitude within 10km of the sea ice edge; yes, cm-scale, long period (30ăˇ AˇT secor longer) waves can penetrate the pack for long distances. Make that clearer.

Noted.

14) Figure 1... One of the image maps needs a scale bar. Check north arrow, also. I find that a few lat-long lines are better for Antarctica anyway, where the direction of north can vary from one side of the image or map to the other.

This will be added to figure 1.

15) Line 208 – '....then decayed on 19 September at 1300 UTC.' Suggest you re-write this with something like, '...then decayed on 19 September, with 9XX hPa central pressure by 1300 UTC.'

Agreed.

16) Line 220-230 – might want to note that the atmospheric river (need to introduce the acronym before using it), in carrying a lot of moisture toward the ice sheet and upward, was inevitably warmed by condensation (released heat) as well as by downward flow, a kind of foehn event.

Noted.

17) Seems like you could eliminate a few of the 8 panels in Figures 2 and 3. Not sure you need to show the anomalies (or perhaps move them to supplemental information)

Noted. We will reduce the number of panels.

18) Line 242–do you have an estimate of the wind speed at an Amery AWS during this period?

Unfortunately, we were not able to find available data during this period at an Amery AWS.

19) Fig. 4g–won't this just be a measure of the sea ice ridging in the region in the aftermath of the intense north-northeasterly first cyclone and first twin cyclone?

We will check on sentinel sensitivity to sea ice presence and revise this figure accordingly. Thanks.

20) Line 251 – 'meandered' is an informal term, and means a sinuous path – is that the case?

We will change 'meandered' to 'moved'.

21) Line 281 –change to ' far-south-reaching blocking highs'.

Done.

22) Line 290 – change to 'advected very cold air (2 m temperatures of ..'

Done.

23) Line293-296–these sentences have redundant information, shorten: the atmospheric river continued to advect large amounts of moisture and very warm air to the ice shelf.

Done.

24) Line 301-317 and elsewhere – The authors frequently use percentiles and standard deviations in the text to describe the extreme conditions, but these are difficult to appreciate without a lot of background awareness of typical weather in the region. While it does convey the unusualness of the conditions, I suggest shifting the description here and elsewhere to -always- using the absolute values and bearings first (rather than u-values and percentiles, or std. dev. numbers) and then put the statistically-based assessments in parentheses after the 'real' numbers. Readers can more easily grasp '25 m/s winds' relative to '99 percentile (5 std dev above mean) winds'.

Thanks for this good suggestion, we will follow this format.

25) Line 335-338 – I don't see how this would be the case. Offshore winds compact and compress the frazil ice into thicker ridged ice, and expose more ocean surface, a kind of sea ice factory. This compressed ice cools and becomes part of the pack, and is generally quite durable because of the wind compression.

Agreed. We will reformulate this statement accordingly.

26) Line 339-345 – I really doubt this – you are showing more than 500 km of mostly 100% pack ice – even a few km of pack ice dramatically reduces the effect of swell! This line of argument must be removed. The small open areas you show in Figure 7 –might develop a bit of chop and swell -in the fetch of the <50% ice cover areas alone- and that's about it.

Agreed. Line will be removed.

27) Line 346-351 – No again! If there was an intense moisture plume in the events (the IVT 'river') then there was intense snowfall over the sea ice, thickening it and toughening it. The new areas of low concentration you show are -local- and due to intense wind compaction in the leeward

direction. And, Lines 349-351, the local waves induce insignificant flexure or vibration of the ice shelf, almost zero, the ice is far too thick, any flexure would be completely elastic.

Noted. This will be revised in the light of the new results on gravitationally-driven calving.

28) Lines 358-368 – authors are on the wrong track. Much of this and the preceding~100 lines are speculation. Lines 369-409–mostly speculation, and wrong direction.... The kind of high-frequency swell generated by high winds in closed near-shore polynyas would have a trivial to non-existent effect on a shelf hundreds of meters thick, cold-ice-cored, like the Amery.

Noted. We have revised this after the new insight on the ocean slope and your major comments above.

---

## Author Response (AR2)

Manuscript TC-2020-219:

**Atmospheric extremes triggered the biggest calving event in more than 50 years at the Amery Ice shelf in September 2019.**

*By D. Francis et al.*

We have changed the title and added a sentence to the abstract to underscore the role of the ocean slope in the calving event as suggested by the reviewer and the editor.

Below is the new abstract and key words.

Many thanks for your feedback and valuable comments.

**Atmospheric extremes caused high oceanward sea-surface slope triggering the biggest calving event in more than 50 years at the Amery Ice shelf.**

**Abstract**

Ice shelf instability is one of the main sources of uncertainty in Antarctica's contribution to future sea level rise. Calving events play a crucial role in ice shelf weakening but remain unpredictable and their governing processes are still poorly understood. In this study, we analyze the unexpected September 2019 calving event from the Amery Ice Shelf, the largest since 1963 and which occurred almost a decade earlier than expected, to better understand the role of the atmosphere in calving. We find that atmospheric extremes provided a deterministic role in this event. A series of anomalously-deep and stationary explosive twin polar cyclones over the Cooperation and Davis Seas which generated tides and wind-driven ocean slope leading to fracture amplification along the pre-existing rift, and ultimately calving of the massive iceberg. The calving was triggered by high oceanward sea-surface slopes produced by the storms. The observed record-anomalous atmospheric conditions were promoted by blocking ridges and Antarctic-wide anomalous poleward transport of heat and moisture. Blocking highs helped in (i) directing moist and warm air masses towards the ice shelf and in (ii) maintaining stationary the observed extreme cyclones at the front of the ice shelf for several days. Accumulation of cold air over the ice sheet, due to the blocking highs, led to the formation of an intense cold-high pressure over the ice sheet, which helped fuel sustained anomalously-deep cyclones via increased baroclinicity. Our results stress the importance of atmospheric extremes in ice shelf dynamics via tides and sea surface slope and the need to be accounted for when considering Antarctic ice shelf variability and contribution to sea level, especially given that more of these extremes are predicted under a warmer climate.

**Keywords:** Ice shelf calving, icebergs, Amery Ice Shelf, East Antarctica, blocking highs, polar cyclones, explosive cyclones, sea surface slope.